# Learning to Weight Parameters for Training Data Attribution

**Shuangqi Li**[1]    **Hieu Le**[1,3]    **Jingyi Xu**[2]    **Mathieu Salzmann**[1]
[1]EPFL, Switzerland    [2]Stony Brook University, USA    [3]UNC Charlotte, USA
`{shuangqi.li, mathieu.salzmann}@epfl.ch`
`jingyixu@cs.stonybrook.edu   hle40@charlotte.edu`

## Abstract

We study gradient-based data attribution, aiming to identify which training examples most influence a given output. Existing methods for this task either treat network parameters uniformly or rely on implicit weighting derived from Hessian approximations, which do not fully model functional heterogeneity of network parameters. To address this, we propose a method to explicitly learn parameter importance weights directly from data, without requiring annotated labels. Our approach improves attribution accuracy across diverse tasks, including image classification, language modeling, and diffusion, and enables fine-grained attribution for concepts like subject and style.

## 1 Introduction

Tracing how specific training examples influence the outputs of generative models—known as data attribution—is critical for transparency, copyright protection, and data governance. Early gradient-based methods (Yeh et al., 2018; Pruthi et al., 2020) offered a practical solution by estimating influence through the direct similarity of parameter gradients. A more theoretically grounded approach was established by Influence Functions (Koh & Liang, 2017), a seminal technique that leverages the Hessian to better approximate how model parameters would change if a training point were up-weighted. While powerful, computing the true Hessian is intractable in practice. This challenge has spurred a variety of approximation strategies, including leveraging Arnoldi iteration to estimate inverse-Hessian-vector products (Schioppa et al., 2022b), using structured approximations of the Hessian such as EK-FAC (Grosse et al., 2023), or building an empirical kernel from projected gradients (Park et al., 2023).

We observe that attribution quality is not uniform across parameters, but instead varies systematically by group. Both theory and practice point to this heterogeneity. From theory, influence functions already suggest that different parameters should matter unequally: the Hessian inverse rescales gradients according to curvature of the loss function, amplifying some directions more than others. In practice, beyond such rescaling, a common strategy is to restrict attribution to a subset of parameters, often to only a classification model's final layer (Koh & Liang, 2017; Pruthi et al., 2020; Yeh et al., 2022), or to the first up-sampling block in a UNet (Lin et al., 2024). This restriction effectively makes ad-hoc choices about which parameter groups contribute to attribution. In this paper, we will further demonstrate this heterogeneous attribution quality in Section 3. For example, we show that in diffusion models, up-block layers achieve higher Linear Datamodeling Scores (LDS) (Park et al., 2023), and self-attention layers often outperform cross-attention layers (Figure 1). These empirical findings are also consistent with common observations that different layers tend to encode different aspects of the output (Ronneberger et al., 2015; Tumanyan et al., 2023; Si et al., 2024).

At the same time, the exact conditions assumed by influence functions are rarely satisfied in large generative models: parameters are typically not at true optima, the Hessian is intractable, and practical methods rely on approximations such as EK-FAC combined with random projections. These approximations offer only indirect and potentially noisy estimates of parameter importance, limiting their reliability. Thus, while both theory and practice suggest that attribution quality varies across parameters, current methods only partially capture this effect, leaving room for approaches that address it more directly.

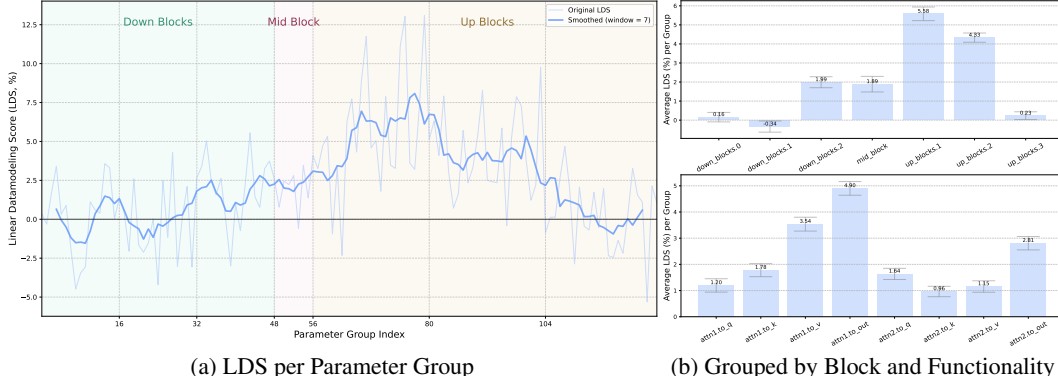

(a) LDS per Parameter Group       (b) Grouped by Block and Functionality

Figure 1: **Attribution Strength Across Parameter Groups** measured using the Linear Datamodeling Score (LDS) (Park et al., 2023). (a) The LDS computed using gradients from individual parameter groups shows significant variations in attribution strength. (b) Aggregating the LDS by block depth (e.g., "down_blocks.0" representing the first downsampling block in the UNet) and functionality within each block (e.g., "attn1" representing the self-attention layers, "attn2.to_q" representing the query projection layers in cross-attention modules) reveals significantly varied attribution strength for different network depths and functional components. Error bars are calculated over 1,000 samples.

To address this, we propose a data-driven method that learns parameter-group weights directly, explicitly aiming to improve attribution quality. Our approach bypasses noisy approximations, adapts across architectures and training regimes, and yields weights that are both effective and interpretable. We cast this into a unified framework that generalizes gradient-based attribution methods, and design a self-supervised objective that learns weights without ground-truth attribution labels. By bootstrapping from the influence rankings produced by existing attribution methods, the method adapts to the heterogeneity of different architectures and training regimes. In doing so, it not only improves attribution accuracy across both image and text generation tasks, but also yields interpretable insights into how distinct parameter groups specialize in aspects of generation such as subject, style, or background.

In summary, our contributions are as follows:

- We show that attribution quality varies systematically across parameter groups, consistent with both theory and practice, highlighting an opportunity to more directly account for this heterogeneity.

- We empirically demonstrate that attribution quality varies significantly across parameter groups in diffusion models, motivating the need for adaptive attribution strategies.

- We propose a unified framework with a self-supervised objective that learns parameter-group weights directly from data.

- Our approach improves attribution accuracy across vision and language tasks, generalizes across architectures and datasets, and provides interpretable semantic disentanglement (e.g., subject vs. style).

## 2 BACKGROUND AND RELATED WORK

**Data Attribution.** The goal of data attribution is to trace a given model output back to its influential training examples. Specifically, a data attribution method $\tau(x, \mathcal{D})$ is a function $\tau : \mathcal{X} \times \mathcal{X}^N \to \mathbb{R}^N$ that, for any sample $x \in \mathcal{X}$ and a training set $\mathcal{D}$ of size $N$, assigns a score to each training example $x^n \in \mathcal{D}$ indicating its contribution to a pre-specified model output function, e.g., the loss function $\mathcal{L}(x; \theta^*(\mathcal{D}))$, where $\theta^*(\mathcal{D})$ represents the model parameters optimized on $\mathcal{D}$.

Various attribution methods have been developed. One line of research consists of sampling-based methods, including empirical influence functions (Feldman & Zhang, 2020), Shapley value estimators (Jia et al., 2019; Ghorbani & Zou, 2019; Covert & Lee, 2021), datamodels (Ilyas et al., 2022), and unlearning-based methods (Wang et al., 2024; Isonuma & Titov, 2024). However, such methods

require training numerous models and are often prohibitively costly for practical-sized models. In contrast, gradient-based methods that exploit the information in the gradient with respect to the model parameters are computationally more favorable. In this context, TracIn (Charpiat et al., 2019; Pruthi et al., 2020) computes the similarity between the gradient from a sample of interest and from the training examples; Influence Functions (Koh & Liang, 2017; Schioppa et al., 2022a; Mlodozeniec et al., 2024) and TRAK (Park et al., 2023) go beyond a simple gradient agreement strategy but introduce computation-intensive matrix inversion terms; TRAK (Park et al., 2023) projects the gradient into a lower-dimensional space to alleviate this issue. In this work, we develop an approach able to improve gradient-based attribution methods.

**Attribution in Diffusion Models and LLMs.** Our experiments extend beyond classification to the frontiers of generative AI, where attribution is critical. In image generation, the rise of powerful diffusion models (Ramesh et al., 2022; Saharia et al., 2022; Rombach et al., 2022) has created an urgent need for attribution to address ethical and privacy concerns (Carlini et al., 2023; Somepalli et al., 2023; Birhane et al., 2021). This has led to specialized methods that adapt attribution to the unique properties of diffusion. Journey TRAK (Georgiev et al., 2023) first extended TRAK to handling the multi-step denoising process of diffusion models. Subsequent work then focused on refining the core gradient signal itself: D-TRAK (Zheng et al., 2023) systematically explored different model output functions for computing the loss gradient, while DAS (Lin et al., 2024) derived a new output function specifically for the diffusion objective. A parallel challenge exists for Large Language Models (LLMs), whose immense scale demands highly efficient attribution. Grosse et al. (2023) scaled influence functions to billion-parameter models using an Eigenvalue-corrected K-FAC approximation of the Hessian. LoGRA (Choe et al., 2024) further improved efficiency with a novel gradient projection strategy. Most recently, TrackStar (Chang et al., 2024) integrated existing innovations, including gradient normalization and Hessian approximation, to achieve robust attribution at scale. Nevertheless, none of the above-mentioned methods explicitly model the fact that different network components connect the model's output and the training samples in different ways.

**Parameter Importance Heterogeneity.** The heterogeneous nature of parameter importance is considered mainly in the field of model pruning (Sun et al., 2023; Ma et al., 2023; Liu et al., 2023; Jaiswal et al., 2023; Chen et al., 2020), which aims to identify and remove redundant parameters to improve model inference efficiency. The parameter importance heterogeneity has also been considered in the field of machine unlearning, knowledge editing, and quantization, in the form of knowledge localization (Yu et al., 2023; Wu et al., 2023; Wang & Veitch, 2025) and mixed-precision optimization (Kim et al., 2023; Cui & Wang, 2024). To the best of our knowledge, our work is the first to explicitly address parameter heterogeneity in data attribution.

## 3 PARAMETER HETEROGENEITY IN DATA ATTRIBUTION

In this section, we review the theoretical basis of gradient-based attribution methods and highlight their implicit assumptions about parameter importance. We then provide empirical evidence that attribution strength is heterogeneous across parameters in diffusion models.

### 3.1 IMPLICIT PARAMETER HETEROGENEITY IN PREVIOUS WORK

Gradient-based attribution methods typically compute the influence of training examples via gradient similarity. A common simplification of TracIn (Charpiat et al., 2019; Pruthi et al., 2020) computes the dot product between the gradient at a query sample ($x_{\text{query}}$) and that at a training point ($x^n$):

$$\tau_{\text{TracIn}}(x_{\text{query}}, x^n) = \nabla_\theta \mathcal{L}(x_{\text{query}}; \theta)^\top \cdot \nabla_\theta \mathcal{L}(x^n; \theta), \tag{1}$$

where $\nabla_\theta \mathcal{L}$ denotes the gradient of the loss function w.r.t. the model parameters $\theta$. This formulation inherently treats all parameters uniformly, assuming equal contribution to the attribution score.

However, this uniform weighting assumption overlooks the functional specialization within complex deep models. For example, in the UNet architecture, different components (e.g., attention vs. convolutional layers, shallow vs. deep layers) serve distinct roles. Deeper layers typically capture more holistic semantics, while shallower layers focus on style and texture (Tumanyan et al., 2023; Si et al., 2024). This structural heterogeneity strongly suggests that gradients originating from different parameter locations carry diverse types and amounts of attribution information, making uniform weighting suboptimal.

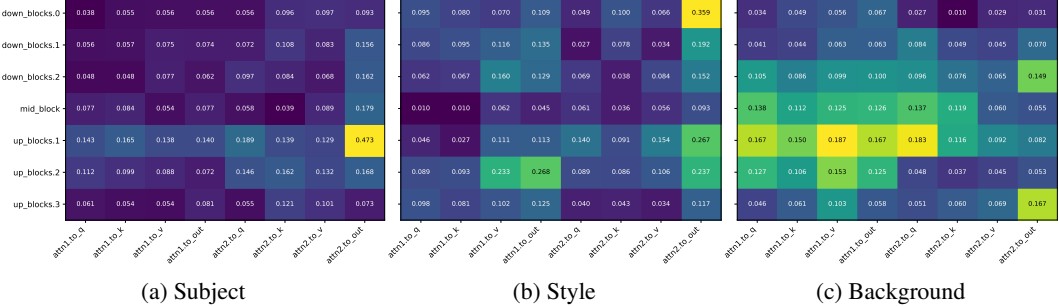

Figure 2: **Parameter specialization by semantic element.** Heatmaps show average recall@10 for attributing (a) subject, (b) style, and (c) background using gradients from different parameter groups (UNet block depth and attention components). Brighter indicates stronger attribution strength for that semantic element by that parameter group.

More sophisticated methods, such as the kernelized approach TRAK (Park et al., 2023), move beyond a simple dot product by incorporating a kernel to mediate the similarity score:

$$\tau_{\text{TRAK}}(x_{\text{query}}, x^n) = \left(P^\top \nabla_\theta \mathcal{L}(x_{\text{query}}; \theta)\right)^\top \cdot \left(\mathbf{\Phi}^\top \mathbf{\Phi} + \lambda I\right)^{-1} \cdot P^\top \nabla_\theta \mathcal{L}(x^n; \theta), \qquad (2)$$

$$\text{where } \mathbf{\Phi} = [\phi(x^1), \dots, \phi(x^N)]^\top, \quad \phi(x^i) = P^\top \nabla_\theta \mathcal{L}(x^i; \theta). \qquad (3)$$

Here $P$ is a random matrix that projects the gradients to a lower-dimensional feature space, making the formulation a scalable analogue to Influence Functions (Koh & Liang, 2017). The term $\left(\mathbf{\Phi}^\top \mathbf{\Phi} + \lambda I\right)^{-1}$ preconditions gradient similarity using second-order structure estimated from the projected gradients. While theoretically motivated, these kernel-based methods face several practical limitations that affect their precision. First, the kernel is an *approximation* of the loss curvature. The formal derivation of influence functions assumes access to the true Hessian, which is intractable for deep models. Existing methods therefore rely on proxies such as the damped Gauss-Newton Hessian (as in TRAK) or the Kronecker-Factored approximation (Grosse et al., 2023), which do not perfectly capture the curvature. Second, the theory assumes that the model has fully converged, whereas the gradients are typically computed from checkpoints that are not at a true local minimum. Finally, the gradient features suffer from the information loss incurred by random projection.

Consequently, the implicit re-weighting effect is derived from an inexact and noisy signal. As such, the resulting implicit weighting may not accurately reflect the true importance of different parameter groups, motivating a direct empirical analysis of the attribution signal's underlying structure.

## 3.2 ATTRIBUTION STRENGTH ACROSS DEPTH AND FUNCTIONAL COMPONENTS

To quantify the variation in attribution signal, we measure the *attribution strength* of different parameter groups. We define attribution strength as the effectiveness of gradients from a parameter group in identifying influential training examples. We quantify attribution using the Linear Datamodeling Score (LDS) (Park et al., 2023), which measures the correlation between a model's sensitivity to training-data subsets and per-sample attribution scores (higher is better; see Appendix B.2.1). To assess how attribution strength varies throughout the network, we fine-tuned a Stable Diffusion model on ArtBench-2 (Liao et al., 2022) using LoRA (Hu et al., 2022) and performed attribution using D-TRAK (Zheng et al., 2023). We partitioned the LoRA parameters into 128 consecutive groups by location within the UNet and computed LDS using only gradients from each group.

Figure 1a shows the LDS for individual parameter groups, revealing substantial variations in attribution performance across the network. Some groups achieve high LDS, while others—particularly in shallower blocks—contribute minimally. Aggregating scores by structural and functional categories further clarifies these trends. Grouping by block depth (Figure 1b, top) highlights dramatic differences, with `up_blocks.1` and `up_blocks.2` exhibiting much higher average LDS (5.58) than others. Grouping by functionality within each architecture block (Figure 1b, bottom) reveals that attribution strength also varies significantly by component type; for instance, the output projection layers (`attn1.to_out`, `attn2.to_out`) consistently show higher average LDS, indicating significantly greater attribution strength compared to other attention components such as cross-attention

key/query/value projections (`attn2.to_k`, `attn2.to_q`, `attn2.to_v`). This confirms that attribution strength varies depending on both parameter location and functional component. Furthermore, as we show via a cosine-similarity analysis in Appendix C.1, the pattern of per-group LDS scores and the learned weights is highly consistent across datasets and attribution methods, indicating that this heterogeneity reflects a stable, intrinsic property of the model rather than an artifact of any specific setting.

### 3.3 ATTRIBUTION STRENGTH BY SEMANTIC ELEMENT

Beyond architectural location, we hypothesize that parameter groups also specialize in attributing distinct semantic elements (e.g., subject, style, background). To investigate this, we synthesized a dataset of 600 images, evenly divided across three attribute types. For each attribute, we defined 10 categories (e.g., *Bulbasaur* for subject, *blacklight painting* for style, *beach* for background) and collected 20 unique images per category. We then fine-tuned a text-to-image Stable Diffusion model on this dataset and generated new images using prompts that combined these elements (e.g., "A [style] of a [subject], [background]"). For each generated output, we treated the 20 training samples that matched its constituent subject, style, or background as the ground-truth contributors for that respective attribute. Accuracy was evaluated using recall@10—the fraction of the top-10 attributed training samples that matched the ground-truth set for a given semantic element.

Figure 2 visualizes the results, confirming a clear semantic specialization. The heatmaps show distinct patterns of attribution strength for subject (a), style (b), and background (c). Different parameter groups exhibit peak attribution strength for different semantic elements; for example, style attribution tends to peak in earlier layers of the network, whereas background attribution is strongest in specific attention components. This provides strong evidence that different network components are indeed responsible for attributing distinct semantic aspects of a generated image.

Taken together, our experiments reveal a clear conclusion: Attribution signal is highly heterogeneous, concentrated in architecturally and semantically distinct components of the network. This underlying structure is a critical source of information that the uniform or implicit weighting schemes of prior methods fail to exploit. This finding motivates a more direct strategy: Explicitly learning parameter importance from data, an approach we introduce below.

## 4 LEARNING TO WEIGHT PARAMETERS FOR DATA ATTRIBUTION

### 4.1 PARAMETER-WEIGHTED DATA ATTRIBUTION: FORMULATION

Let the model parameters $\theta$ be partitioned into $M$ disjoint groups, $\theta = \{\theta_1, \ldots, \theta_M\}$, such as individual tensors or layers. For an input $x$, let $g_j(x)$ denote the gradient-derived feature vector for parameter group $\theta_j$. Attribution methods typically use the concatenated gradient features $g(x) = [g_1(x), \ldots, g_M(x)]$ to compare a query example $x_{\text{query}}$ with a training example $x^n$.

We introduce a learnable, non-negative weight vector $w = \{w_1, \ldots, w_M\}$, where each $w_j$ scales the contribution of the corresponding parameter group. This approach is motivated by the insight that different parameter groups contribute attribution signals of varying quality. The reweighted query feature is thus

$$\tilde{g}(x; w) = [w_1 g_1(x), \ldots, w_M g_M(x)]. \tag{4}$$

Let $\text{Diag}(w)$ denote the diagonal matrix that scales the coordinates of $g(x)$ by the corresponding group weights, i.e., it repeats $w_j$ across all dimensions of $g_j(x)$, so that $\tilde{g}(x; w) = \text{Diag}(w) \, g(x)$. Then, the **parameter-weighted attribution score** between $x_{\text{query}}$ and $x^n$ is

$$\tilde{\tau}(x_{\text{query}}, x^n; w) = g(x_{\text{query}})^\top \cdot \text{Diag}(w) \cdot K \cdot g(x^n), \tag{5}$$

where the matrix $K$ defines a similarity metric. This formulation subsumes both simple and kernelized methods: for TracIn-style attribution, $K$ is the identity matrix, which yields a weighted dot product (cf. Eq. 1); for kernelized methods such as TRAK and its variants, $K$ is a kernel matrix, for example the one built from projected features, $(\Phi^\top \Phi + \lambda I)^{-1}$. For the special case $K = I$, weighting either the query or the training feature (or both) is equivalent up to a reparameterization of $w$, since a global scale on one side can be absorbed into the weights. For general kernels, however, the training-side term $K \, g(x^n)$ is precomputed once for all training examples to make attribution scalable. Applying

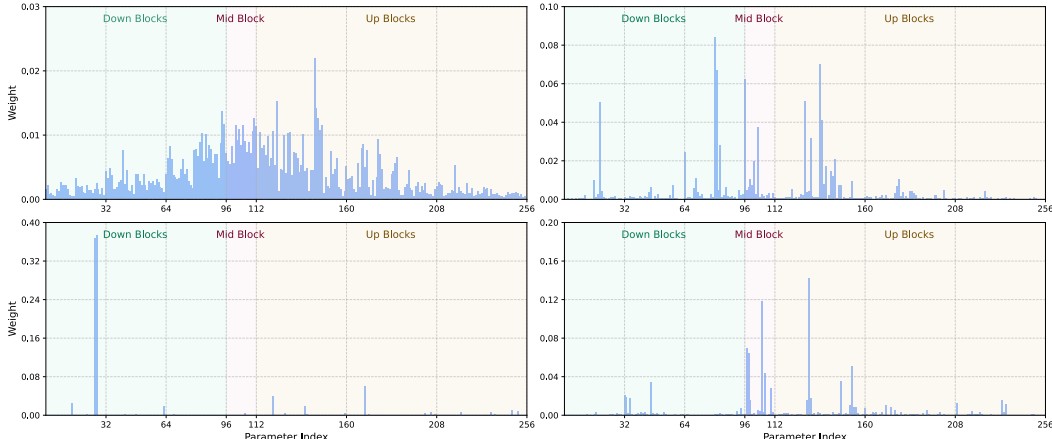

Figure 3: **Learned Parameter Importance Weights** for overall (top-left), subject (top-right), style (bottom-left), and background (bottom-right) attribution. Each plot shows the learned weights across 256 parameter groups in the UNet, organized by Down, Mid, and Up blocks, with different colors indicating each block type.

weights symmetrically would require recomputing this term every time $w$ is updated, which would make weight learning prohibitively expensive. We therefore apply $\text{Diag}(w)$ only to the query features and treat the training-side features $K \, g(x^n)$ as fixed. Conceptually, $w$ specifies how much we trust each parameter group when reading the query's gradient signal.

This unified framework allows any gradient-based attribution method to benefit from learned parameter importance weights, improving both attribution performance and interpretability.

## 4.2 SELF-SUPERVISED WEIGHT LEARNING

Our goal is to learn weighting parameters $w$ that maximize data attribution quality. Obtaining ground-truth attribution scores for direct supervision is generally infeasible. We therefore propose a self-supervised approach that bootstraps from the rankings of an existing attribution method. The core assumption is that the top-$k$ scoring training examples, according to a base attribution method, can serve as *pseudo ground-truth positives*. While the initial ranking is imperfect, it provides a weak supervisory signal that the optimization can refine.

Formally, for a query $x_{\text{query}}$ and training set $\mathcal{D}$, let $\widetilde{\tau}(x_{\text{query}}, \mathcal{D}; w) \in \mathbb{R}^N$ be the vector of all parameter-weighted attribution scores. Let $\mathcal{I}_{\text{top-k}}(w) \subset \{1, \ldots, N\}$ denote the set of indices corresponding to the top $k$ scores in this vector. We optimize the weights by maximizing the average score of these pseudo-positives, normalized by the overall score magnitude. This is expressed as the loss function:

$$\mathcal{L}_{\text{SSL}}(w; x_{\text{query}}, \mathcal{D}, k) = -\frac{1}{\|\widetilde{\tau}(x_{\text{query}}, \mathcal{D}; w)\|_2} \left( \frac{1}{k} \sum_{i \in \mathcal{I}_{\text{top-k}}(w)} \widetilde{\tau}(x_{\text{query}}, x^i; w) \right). \quad (6)$$

During optimization, the set of top-$k$ indices, $\mathcal{I}_{\text{top-k}}(w)$, is re-evaluated with the updated weights at each step, allowing the model to bootstrap a progressively better signal.

As we formally derive in Appendix A, minimizing this loss serves as a principled proxy for **maximizing the SNR of the attribution score**. Specifically, our weighted formulation (Eq. 5) is designed such that the numerator acts as an estimate of the signal strength, while the denominator, the $\ell_2$ norm, estimates the total noise level.

To learn a single, generalizable weight vector $w$, we minimize the expectation of this loss over a distribution of query samples $\mathcal{Q}$, adding a regularization term to prevent overfitting. This yields:

$$w^* = \arg\min_{w \geq 0} \mathbb{E}_{x_{\text{query}} \sim \mathcal{Q}}[\mathcal{L}_{\text{SSL}}(w; x_{\text{query}}, \mathcal{D}, k)] + \lambda \|w\|_2. \quad (7)$$

In practice, we enforce weight non-negativity via a softmax parameterization. While performance depends on the hyperparameter $k$, we find it to be stable across a broad range of values (see

Table 1: **LDS (%) on ImageNet classification.** Our method significantly improves attribution for both CNNs and Transformers. We report LDS $\pm$ 95% CI half-width.

| Method | ResNet-18 | | ViT-B/16 | |
| --- | --- | --- | --- | --- |
| | w/o $w$ | $w$ | w/o $w$ | $w$ |
| TracIn | 11.39 $\pm 1.06$ | 23.92 $\pm 1.26$ | 9.67 $\pm 0.94$ | 17.63 $\pm 1.08$ |
| TRAK | 16.86 $\pm 1.03$ | 23.30 $\pm 1.18$ | 14.77 $\pm 0.97$ | 16.74 $\pm 1.03$ |

Table 2: **LDS (%) on WikiText-103 with GPT-2-small.**

| Method | w/o $w$ | $w$ |
| --- | --- | --- |
| TracIn | 6.31 $\pm 0.88$ | 9.23 $\pm 0.96$ |
| TRAK | 12.69 $\pm 1.00$ | 14.63 $\pm 1.04$ |
| LoGRA | 11.42 $\pm 1.00$ | 12.86 $\pm 1.04$ |
| EKFAC [1] | 15.14 $\pm 1.04$ | 18.33 $\pm 1.13$ |

Table 3: **AUC for Mislabeled Data Detection on ImageNet.** We treat the manually mislabeled samples as the positive class and correctly labeled points as negative. Higher is better.

| Method | ResNet-18 | | ViT-B/16 | |
| --- | --- | --- | --- | --- |
| | w/o $w$ | $w$ | w/o $w$ | $w$ |
| TracIn | 54.40 | 61.46 | 71.27 | 83.58 |
| TRAK | 59.66 | 67.72 | 80.08 | 83.48 |

Table 4: **Tail-patch score (%) on WikiText-103.** $\Delta$ denotes the average change per query with 95% CI half-width. Higher is better.

| Method | w/o $w$ | $w$ | $(\Delta)$ |
| --- | --- | --- | --- |
| Random | 0.39 | — | — |
| TracIn | 4.66 | 5.60 | 0.94 $\pm 0.17$ |
| TRAK | 7.77 | 7.88 | 0.10 $\pm 0.05$ |
| LoGRA | 6.34 | 6.54 | 0.20 $\pm 0.09$ |
| EKFAC | 5.54 | 6.09 | 0.55 $\pm 0.11$ |

Appendix C.2 for a sensitivity analysis). We have also experimented with other loss functions (see Appendix C.4), such as a supervised loss using pseudo labels resulting from data augmentation, but found them inferior compared to our self-supervised one (Eq. 6). The weight learning process is highly efficient, typically converging in under one minute. This efficiency stems from two factors: First, we only learn a small number of weights (one per parameter group, e.g., one per layer); second, our formulation allows us to precompute attribution contributions from each parameter group, and then apply weights to the group-wise attribution scores during optimization, avoiding recomputing attribution scores with weighted gradient features each iteration.

**Fine-Grained Attribution.** Beyond general attribution, we can aim to isolate the influence of training data on specific aspects of the generated content, such as style or subject. Our data-driven approach naturally extends to such fine-grained attribution tasks by learning specialized weight sets for different semantic aspects (e.g., $w_{style}$, $w_{subject}$, $w_{background}$). The key insight is to construct query sets $\mathcal{Q}$ that emphasize the target attribute while leaving other aspects unspecified. For instance, to learn style-specific weights, we generate query images by specifying different styles in the prompts while not specifying subjects or backgrounds. This causes style-relevant training samples to rank higher in the attribution scores, encouraging the optimization to up-weight parameter groups that consistently contribute to style semantics. Figure 3 shows that these specialized weights are more focused than general-purpose ones, with distinct patterns for different semantic elements.

## 5 EXPERIMENTS

In this section, we demonstrate the effectiveness of our proposed framework. We first show that learning parameter importance consistently improves standard data attribution across diverse domains, including image classification, language modeling, and image generation. We then show that our method can be extended to perform fine-grained, semantic-specific attribution. Further details on all models, datasets, baseline methods, and evaluations are provided in Appendices B.1, B.3, and B.2.

### 5.1 STANDARD DATA ATTRIBUTION

We apply our self-supervised method to learn parameter weights for a suite of prominent gradient-based attribution methods, including TracIn (Pruthi et al., 2020), TRAK (Park et al., 2023), EKFAC (Grosse et al., 2023), JourneyTRAK (Georgiev et al., 2023), D-TRAK (Zheng et al., 2023), and DAS (Lin et al., 2024). Across all experiments, we enforce **three disjoint roles** for data: (i) *model training data* for learning parameters $\theta$, (ii) *weight-learning data* for optimizing parameter-group weights $w$, and (iii) *evaluation data* for LDS, tail-patch, and fine-grained Recall@10. The precise

---

[1] We use the EKFAC implementation in the LoGRA codebase.

Table 5: **LDS (%) comparison on diffusion models.** Our method improves performance across multiple datasets and attribution baselines. We report LDS $\pm$ 95% CI half-width.

| Method | ArtBench-2 | | Naruto | | SB-Pokemon | | CIFAR-2 | |
|--------|-----------|-----------|-----------|-----------|-----------|-----------|-----------|-----------|
| | w/o $w$ | $w$ | w/o $w$ | $w$ | w/o $w$ | $w$ | w/o $w$ | $w$ |
| TracIn | $17.63_{\pm0.96}$ | $22.02_{\pm1.05}$ | $10.54_{\pm0.82}$ | $13.59_{\pm0.81}$ | $9.34_{\pm0.93}$ | $11.79_{\pm0.91}$ | $1.39_{\pm0.78}$ | $8.48_{\pm0.79}$ |
| TRAK | $18.39_{\pm0.96}$ | $22.15_{\pm0.99}$ | $14.61_{\pm0.80}$ | $17.02_{\pm0.83}$ | $10.68_{\pm0.92}$ | $12.24_{\pm0.89}$ | $8.51_{\pm0.78}$ | $10.59_{\pm0.78}$ |
| JTRAK | $11.56_{\pm0.89}$ | $16.00_{\pm1.00}$ | $13.41_{\pm0.80}$ | $14.56_{\pm0.82}$ | $7.97_{\pm0.93}$ | $9.24_{\pm0.94}$ | $6.03_{\pm0.79}$ | $8.39_{\pm0.79}$ |
| DTRAK | $22.72_{\pm1.02}$ | $25.15_{\pm1.09}$ | $16.75_{\pm0.85}$ | $17.85_{\pm0.85}$ | $33.88_{\pm0.91}$ | $35.05_{\pm0.92}$ | $10.17_{\pm0.78}$ | $12.18_{\pm0.77}$ |
| DAS | $30.47_{\pm1.15}$ | $31.58_{\pm1.15}$ | $18.72_{\pm0.92}$ | $20.44_{\pm0.92}$ | $33.55_{\pm1.04}$ | $36.12_{\pm0.99}$ | $12.66_{\pm0.80}$ | $13.79_{\pm0.81}$ |

splits for each setting are described below and summarized in Table 7 in Appendix B.1. We assess the impact of our method across multiple evaluation metrics, detailed in the following subsections.

### 5.1.1 IMAGE CLASSIFICATION

We first evaluate our method on image classification by training a ResNet-18 (He et al., 2016) and a ViT-B/16 (Dosovitskiy et al., 2020) model on a 50,000-sample subset of the ImageNet (Deng et al., 2009) training split. Weights $w$ are learned using 1,000 additional images sampled from the ImageNet training split, disjoint from this 50,000-sample subset. LDS (Park et al., 2023) is evaluated on 1,000 images sampled from the ImageNet validation split, disjoint from both the model-training and weight-learning data. As shown in Table 1, our method brings substantial improvements to both TracIn and TRAK across both architectures.

To assess performance on a practical downstream task, we evaluate on mislabeled data detection. The key insight, following prior work (Koh & Liang, 2017; Pruthi et al., 2020), is that mislabeled examples, being outliers, tend to exhibit high self-influence scores. We therefore rank all training data by normalized self-influence. By treating mislabeled points as the positive class and correctly labeled points as the negative class, we can measure the ability to separate them by computing the Area Under the ROC Curve (AUC). For this evaluation, we corrupted 10% of the training labels, and compute self-influence on a model trained on the corrupted data. As shown in Table 3, our method consistently improves AUC, indicating that our weighted approach provides a more effective ranking for prioritizing noisy samples for human inspection.

### 5.1.2 LANGUAGE MODELING

We next test our method on language modeling. We train a GPT-2-small (Radford et al., 2019) model from scratch on the WikiText-103 (Merity et al., 2016) training split, containing 232,585 sequences of length 512. Weights $w$ are learned using 523 sequences from the WikiText-103 test split, and LDS is evaluated on 487 sequences from the validation split. Table 2 shows consistent LDS improvements for TracIn, TRAK, LoGRA (Choe et al., 2024), and EKFAC on this held-out validation set.

In addition, we evaluate using the tail-patch score (Chang et al., 2024), a metric designed to measure the direct, additive contribution of helpful training examples. For each query, the procedure is as follows: we identify the top-128 training proponents, treat them as a single batch, and perform one incremental training step (a "tail-patch") on the final model checkpoint using this batch. The score is the resulting increase in the query's target sequence probability, averaged over all queries in the test set. A higher score indicates a better ability to retrieve genuinely useful training data. As shown in Table 4, our weighted approach more effectively identifies these helpful training examples, leading to larger performance gains.

### 5.1.3 IMAGE GENERATION

Finally, we evaluate on diffusion models with attribution methods such as JourneyTRAK (Georgiev et al., 2023), D-TRAK (Zheng et al., 2023), and DAS (Lin et al., 2024), in addition to TracIn and TRAK. We use four datasets: **CIFAR-2**, a subset of CIFAR-10 (Krizhevsky et al., 2009); **ArtBench-2**, a subset of ArtBench (Liao et al., 2022); **Naruto**, a curated dataset of the naruto anime images (Cervenka, 2022); and our synthetic **SB-Pokemon** dataset. We train a DDPM (Ho et al., 2020) for CIFAR-2 and fine-tune Stable Diffusion (Rombach et al., 2022) with LoRA (Hu et al., 2022) for the others using the respective training sets. For each dataset, we learn weights $w$ from a

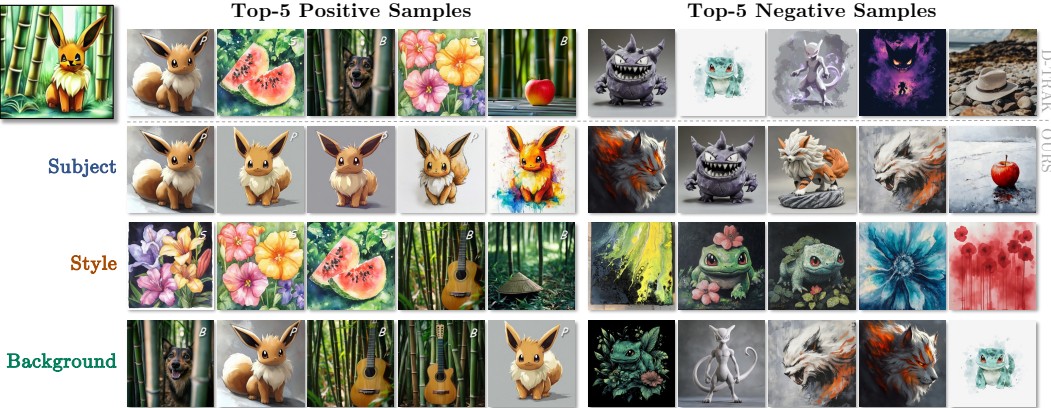

Figure 4: **Fine-grained attribution example.** The query image (top-left) was generated with the prompt "A watercolor illustration of Eevee, in a bamboo forest". The first row shows the top-5 positive and negative training samples identified by standard D-TRAK without weights. Each subsequent row (Subject, Style, Background) displays training samples retrieved using specialized weights learned for that semantic element. Images marked 'S', 'P', or 'B' are ground-truth contributors for style, subject, or background, respectively.

Table 6: **Fine-grained attribution on SB-Pokemon using D-TRAK.** Each row corresponds to using weights learned for a specific semantic target. Columns show Recall@10 (%) for attributing Subject, Style, Background elements, and their summation (All), on the train and validation prompt splits. "Rep. Dist." represents a simple baseline that ranks training images by $\ell_2$ distance to the query in the feature space of a ResNet-50 model pre-trained on ImageNet using MoCo (He et al., 2019).

| | Train | | | | Validation | | | |
|---|---|---|---|---|---|---|---|---|
| Weight target | Subj. | Style | Backg. | All | Subj. | Style | Backg. | All |
| None (w/o $w$) | 22.1 | 64.9 | 9.1 | 96.0 | 19.4 | 61.8 | 17.5 | 98.7 |
| Rep. Dist. (Singla et al., 2023) | 22.0 | 10.5 | 5.7 | 38.2 | 13.4 | 13.4 | 16.3 | 43.1 |
| Subject | **51.6** | 35.5 | 10.9 | 98.0 | **50.6** | 30.6 | 18.4 | **99.6** |
| Style | 17.0 | **82.1** | 0.6 | **99.8** | 15.2 | **73.0** | 10.9 | 99.0 |
| Background | 30.0 | 48.5 | **12.9** | 91.4 | 29.8 | 41.4 | **26.6** | 97.7 |

separate set of model-generated query images, and evaluate LDS on another disjoint set of generated queries, following the prompting strategies described in Appendix B.2 and summarized in Table 7. Table 5 shows that our method significantly improves LDS scores across all methods and datasets.

Across all three domains, our method consistently improves the performance of all data attribution techniques. Furthermore, as shown in Appendix C.5, the learned weights exhibit strong robustness, generalizing well across datasets and attribution methods, suggesting they capture the model's intrinsic characteristics instead of overfitting to specific queries or methods.

## 5.2 FINE-GRAINED DATA ATTRIBUTION EVALUATION

In addition to improving general attribution, our framework enables fine-grained attribution—allowing us to find the influence of training data on specific aspects of an image, such as style, subject, or background. This is done using the same self-supervised method, but with query sets curated to focus on each target attribute (see Section 4.2).

We use our synthetic SB-Pokemon dataset for this experiment. For each semantic element, we learn a separate set of weights: $w_{\text{style}}$, $w_{\text{subject}}$, and $w_{\text{background}}$. The query set $\mathcal{Q}$ for learning each specialized weight set is constructed from images generated using prompts that focus on variations of that specific element while keeping others general. For instance, to learn $w_{\text{background}}$, $\mathcal{Q}$ consists of 200 images generated with prompts emphasizing different backgrounds (e.g., "*in a misty marsh*", "*on a rocky beach*"), without specifying subjects or styles. To assess generalization, we split the 10 categories

for each semantic element into a train split (5 categories) and a validation split (the remaining 5 categories). The prompts used to learn specialized weights contain only concepts from the train split.

For evaluation, we measure Recall@10 (the proportion of the top-10 attributed samples that were known ground-truth contributors) for each semantic element separately. We generate 500 test images for both the train and validation splits. These images are created using prompts that combine elements from the respective splits, following the pattern "A [style] of [subject], [background]". (5 styles $\times$ 5 subjects $\times$ 5 backgrounds $\times$ 4 repeats $= 500$ images).

As shown in Table 6, applying specialized weights improves attribution for the targeted element. The validation results show strong generalization to unseen styles, subjects, and backgrounds. Figure 4 provides qualitative examples supporting the effectiveness of this approach: while some retrieved training images still reflect other concepts, target-concept contributors become much more prominent than under the unweighted baseline, which is expected since reweighting can suppress but not completely remove off-target signal.

## 6 CONCLUSION

We have shown that explicitly learning parameter importance consistently improves gradient-based data attribution. Our self-supervised, SNR-grounded framework boosts attribution accuracy across image classification, language modeling, and diffusion models, and enables fine-grained semantic analysis.

Despite these advantages, our approach has two main limitations. First, the weights are layer-wise in practice, which is computationally efficient but coarse; learning more granular, parameter-wise weights is a natural next step, but may face stronger overfitting. Second, because the self-supervised objective bootstraps from existing attribution methods (for example, TRAK or D-TRAK) and a constructed query dataset, the learned weights inevitably inherit part of their inductive biases. We view this inherited bias as an acceptable trade-off for improving practical attribution quality without requiring ground-truth influence labels.

Overall, our findings demonstrate that parameter heterogeneity can be modeled and learned from data, enabling more accurate, controllable, and interpretable training data attribution.

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

# A    SNR-BASED DERIVATION OF THE SELF-SUPERVISED LOSS

In this section, we provide a theoretical justification for our proposed parameter weighting framework and self-supervised loss, grounding it in Signal-to-Noise Ratio (SNR) maximization principles.

## A.1    A SIGNAL-PLUS-NOISE MODEL FOR ATTRIBUTION

A conventional gradient-based attribution score, such as TracIn, can be decomposed into a sum of scores from $M$ disjoint parameter groups:

$$\tau(x_q, x_n) = \nabla_\theta \mathcal{L}_q^\top \nabla_\theta \mathcal{L}_n = \sum_{j=1}^{M} \nabla_{\theta_j} \mathcal{L}_q^\top \nabla_{\theta_j} \mathcal{L}_n \equiv \sum_{j=1}^{M} a_j(x_q, x_n).$$

This formulation implicitly assigns a uniform weight of 1 to each per-group score $a_j$. Our method generalizes this to a weighted sum by introducing learnable, non-negative weights $w_j$:

$$\widetilde{\tau}(x_q, x_n; w) = \sum_{j=1}^{M} w_j a_j(x_q, x_n).$$

To formalize the rationale for this, we model the per-group score $a_j$ as a noisy linear measurement of a true, unknown influence signal $I(x_q, x_n) \in \mathbb{R}$:

$$a_j(x_q, x_n) = \alpha_j \cdot I(x_q, x_n) + \epsilon_j,$$

where $\alpha_j \geq 0$ is the group's signal sensitivity and $\epsilon_j$ is zero-mean random noise with variance $\sigma_j^2$, assumed to be independent across groups. The weighted attribution score for a single pair is then:

$$\widetilde{\tau}(x_q, x_n; w) = \left( \sum_{j=1}^{M} w_j \alpha_j \right) I(x_q, x_n) + \sum_{j=1}^{M} w_j \epsilon_j.$$

The quality of this aggregated score can be measured by its Signal-to-Noise Ratio (SNR), defined as the ratio of the variance of the signal component to the variance of the noise component:

$$\mathrm{SNR}_w = \frac{\mathrm{Var}[(\sum_j w_j \alpha_j) I]}{\mathrm{Var}[\sum_j w_j \epsilon_j]} = \frac{(\sum_j w_j \alpha_j)^2 \mathrm{Var}(I)}{\sum_j w_j^2 \sigma_j^2}.$$

Since $\mathrm{Var}(I)$ is a constant, maximizing the SNR is equivalent to maximizing $\frac{(\sum_j w_j \alpha_j)^2}{\sum_j w_j^2 \sigma_j^2}$. By the Cauchy-Schwarz inequality, this expression is maximized when the weights are proportional to the group's intrinsic signal-to-noise characteristic: $w_j^* \propto \alpha_j / \sigma_j^2$. This shows that an optimal non-uniform weighting exists and provides a clear theoretical motivation for learning weights.

## A.2    OUR SELF-SUPERVISED LOSS AS A PROXY FOR SNR MAXIMIZATION

Having established a theoretical goal (maximize SNR), we now show that our practical self-supervised objective is a principled proxy for this goal. Let $\widetilde{\tau}(w) \in \mathbb{R}^N$ be the full vector of attribution scores for a given query, and let $\mathcal{I}_{\text{top-k}}(w)$ be the set of indices of the top $k$ scores in that vector. Our objective is to maximize:

$$\mathcal{L}(w) = \frac{\frac{1}{k} \sum_{i \in \mathcal{I}_{\text{top-k}}(w)} \widetilde{\tau}(x_q, x^i; w)}{\|\widetilde{\tau}(w)\|_2}$$

We now demonstrate that maximizing this objective approximates maximizing $\sqrt{\mathrm{SNR}_w}$.

**Numerator Maximizes Signal Sensitivity.** The numerator is the average score of the top-$k$ training examples. Let $\bar{I}_{\text{top-k}}(w) = \mathbb{E}[I(x_q, x_n)|n \in \mathcal{I}_{\text{top-k}}(w)]$ be the average **true** influence for this group. The expected value of the numerator is:

$$\mathbb{E}\left[\frac{1}{k} \sum_{i \in \mathcal{I}_{\text{top-k}}(w)} \widetilde{\tau}(x_q, x^i; w)\right] = \frac{1}{k} \sum_{i \in \mathcal{I}_{\text{top-k}}(w)} \mathbb{E}[\widetilde{\tau}(x_q, x^i; w)]$$

$$= \left(\sum_{j=1}^{M} w_j \alpha_j\right) \left(\frac{1}{k} \sum_{i \in \mathcal{I}_{\text{top-k}}(w)} I(x_q, x^i)\right)$$

$$\approx \left(\sum_{j=1}^{M} w_j \alpha_j\right) \bar{I}_{\text{top-k}}(w).$$

During optimization, both terms on the right-hand side are affected by $w$. As $w$ improves, the ranking becomes more accurate, likely causing $\bar{I}_{\text{top-k}}(w)$ to increase. However, the primary driving force of the optimization is the direct, linear dependence on the effective signal sensitivity term, $\sum_j w_j \alpha_j$. Therefore, maximizing the numerator is primarily driven by finding weights that maximize this term.

**Denominator Estimates Noise Level.** The denominator is the $l_2$ norm of the score vector. Its squared value is $\|\widetilde{\boldsymbol{\tau}}(w)\|_2^2 = \sum_{n=1}^{N} \widetilde{\tau}(x_q, x^n; w)^2$. We assume the vast majority of training examples have negligible true influence ($I(x_q, x^n) \approx 0$). For these examples, the score is dominated by noise: $\widetilde{\tau}(x_q, x^n; w) \approx \sum_j w_j \epsilon_j$. The expected squared norm is therefore dominated by the noise variance:

$$\mathbb{E}[\|\widetilde{\boldsymbol{\tau}}(w)\|_2^2] = \sum_{n=1}^{N} \mathbb{E}[\widetilde{\tau}(x_q, x^n; w)^2] \approx \sum_{n=1}^{N} \mathbb{E}\left[\left(\sum_j w_j \epsilon_j\right)^2\right] = N \sum_j w_j^2 \sigma_j^2.$$

Thus, the denominator $\|\widetilde{\boldsymbol{\tau}}(w)\|_2$ serves as a data-driven estimator proportional to the square root of the total noise power, $\sqrt{\sum_j w_j^2 \sigma_j^2}$.

**Conclusion.** Combining these insights, our practical objective is a direct proxy for the ideal one:

$$\mathcal{L}(w) \propto \frac{\sum_j w_j \alpha_j}{\sqrt{\sum_j w_j^2 \sigma_j^2}} \propto \sqrt{\text{SNR}_w}.$$

This proves that optimizing our proposed loss is a principled method for maximizing the SNR of the attribution score under our signal-plus-noise model. The iterative learning process allows the system to bootstrap its way to a progressively better estimate of the optimal weights $w_j^* \propto \alpha_j / \sigma_j^2$.

**Remark on "metric hacking" LDS.** Our SNR-based objective operates only on the raw attribution scores $\widetilde{\tau}(x_q, x_n; w)$: it increases a top-$k$ average while controlling the overall $\ell_2$ scale, and LDS values (a Spearman correlation over retrained models) never enter the optimization. Furthermore, the same learned weights improve several evaluations that are not defined in terms of LDS (mislabeled-data AUC, tail-patch, and qualitative analyses), and ablations such as removing the normalization term or replacing the Top-$k$ term by a Top-$k$ minus Bottom-$k$ variant either collapse performance or fail to match our main loss. Together, these observations are hard to reconcile with superficial metric hacking and are instead consistent with the intended SNR interpretation, where the loss encourages the attribution mechanism to place more weight on parameter groups whose scores carry stable influence signal and to down-weight groups whose scores behave like noise, rather than manipulating evaluation metrics directly.

Table 7: **Data usage across experiments.** For each setting we distinguish three disjoint roles for data: model training (for learning $\theta$), weight learning (for optimizing $w$), and evaluation (for LDS, tail-patch, and/or fine-grained Recall@10).

| Setting | Model training data | Weight-learning data | Evaluation data |
|---|---|---|---|
| Image classification (ImageNet; ResNet-18 / ViT-B/16) | 50,000 images from the ImageNet training split (10% label corruption applied only for the mislabeled-data experiments) | 1,000 additional images from the ImageNet training split, disjoint from the 50,000-sample subset | 1,000 images from the ImageNet validation split for LDS evaluation; 5,000 training images with corrupted labels for mislabeled data detection experiments. |
| Language modeling (WikiText-103; GPT-2-small) | Texts from the training split concatenated and segmented into 232,585 sequences of length 512 | 523 sequences from the test split | 487 sequences from the validation split (used for LDS and as tail-patch queries) |
| Diffusion models (CIFAR-2, ArtBench-2, Naruto, SB-Pokemon) | Dataset-specific training sets described below | 1,000 model-generated query images per dataset for learning $w$ (for fine-grained SB-Pokemon weights, 200 generated queries containing only train-split concepts) | 1,000 model-generated query images per dataset with disjoint random seeds for LDS; for fine-grained Recall@10 on SB-Pokemon, 500 generated queries per split (train / validation) |

## B    Technical Details

### B.1    Dataset and Model Details

**CIFAR-2 ($32 \times 32$)** The CIFAR-10 (Krizhevsky et al., 2009) dataset consists of 50,000 training and 10,000 test images of size $32 \times 32$ across 10 classes. CIFAR-2 is built by randomly selecting 5,000 samples from the "automobile" and "horse" classes for training, each with 2,500 images.

For the CIFAR-2 dataset, our unconditional DDPM (Ho et al., 2020) comprises 35.7M parameters. The diffusion process operates with a maximum timestep of $T = 1000$, utilizing a linear variance schedule where $\beta$ ranges from $\beta_1 = 10^{-4}$ to $\beta_T = 0.02$. Training incorporates a dropout rate of $0.0$ and random horizontal flip data augmentation. Optimization is performed using AdamW (Loshchilov & Hutter, 2017) with a weight decay of $10^{-6}$. The model is trained over 200 epochs using a batch size of 128. The learning rate follows a cosine annealing schedule, initiated at $10^{-4}$ after a 0.1 fraction warmup period. For inference, we employ the 50-step DDIM solver (Song et al., 2020).

**ArtBench-2 ($256 \times 256$)** ArtBench (Liao et al., 2022) is a dataset of artwork images consisting of 10 distinct artistic styles, each consisting of 5,000 images in the training set. We follow the setup in (Zheng et al., 2023) to build ArtBench-2, a subset of ArtBench, containing 5,000 randomly selected samples from the "post-impressionism" and "ukiyo-e" styles, each with 2,500 training images.

On ArtBench-2, we fine-tune a text-to-image Stable Diffusion model (Rombach et al., 2022) using LoRA (Hu et al., 2022) with the rank set to 128, leading to 25.5M learnable parameters. To accommodate ArtBench's 256×256 resolution, we utilize a Stable Diffusion checkpoint pre-adapted for this size [2] (from an original 512×512 resolution). The model is trained class-conditionally, with textual prompts formatted as "a class painting" (e.g., "a post-impressionism painting"). Training incorporates a dropout rate of 0.0, AdamW optimization (Loshchilov & Hutter, 2017) with $10^{-6}$ weight decay, and random horizontal flip augmentation. We train for 100 epochs with a batch size of 64. The learning rate, initially $3 \times 10^{-4}$, follows a cosine annealing schedule with a 0.1 fraction

---

[2]https://huggingface.co/lambdalabs/miniSD-diffusers

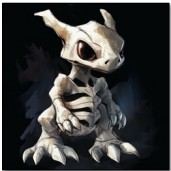 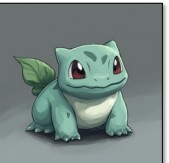 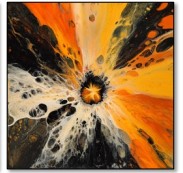 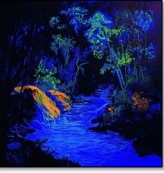 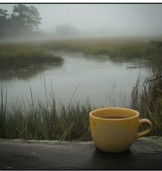 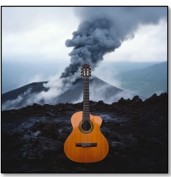

Cubone, black background    Bulbasaur, gray background    An acrylic pour painting    A blacklight painting    a cup, in a misty marsh    a guitar, near a smoking volcano

Figure 5: Illustrative examples from the SB-Pokemon dataset, showcasing samples of Pokemon subjects, artistic styles, and environmental backgrounds. These images were generated using MidJourney V6.1 from the categories listed in Table 8.

Table 8: The 30 categories used for generating the SB-Pokemon dataset, divided into Pokemon subjects, artistic styles, and environmental backgrounds. For each column, the first five categories form the training split, and the subsequent five form the validation split, separated by a dashed line.

| Subjects | Styles | Backgrounds |
|---|---|---|
| Cubone | An acrylic pour painting | in a misty marsh |
| Jigglypuff | A blacklight painting | on a rocky beach |
| Mewtwo | A risograph print | in a sunny meadow |
| Bulbasaur | A cyanotype style photo | in a sunflower field |
| Lickitung | A pointillism painting | near a smoking volcano |
| Rhydon | A watercolor illustration | by a calm lake |
| Gengar | A Gongbi style painting | under aurora borealis |
| Snorlax | A sepia tone photo | on a frozen tundra |
| Arcanine | A red wash painting | in a bamboo forest |
| Eevee | A charcoal pencil sketch | beneath a waterfall |

warmup. For inference, images are generated using a 50-step DDIM solver (Song et al., 2020) and a guidance scale of 7.5 (Ho & Salimans, 2022).

**SB-Pokemon (256 × 256)** The SB-Pokemon dataset was synthetically created using MidJourney V6.1 to evaluate attribution for distinct semantic elements: subject, style, and background. It consists of 600 images, derived from 10 categories each for Pokemon subjects (e.g., Bulbasaur), artistic styles (e.g., blacklight painting), and environmental backgrounds (e.g., beach), with 20 unique images per category. This structured design provides unambiguous ground-truth contributors for fine-grained attribution. The complete list of 30 categories are provided in Table 8, with illustrative examples shown in Figure 5.

We use the same fine-tuning setup as for ArtBench-2.

**Naruto (256 × 256)** We curated the original Naruto dataset (Cervenka, 2022) by replacing the BLIP-generated (Li et al., 2022) captions with more accurate ones generated by GPT-4o (Hurst et al., 2024). For examples, the original incorrect caption "a man with a red hair and a black shirt" was replaced with "a ninja wearing a unique mask with a playful design and spiky brown hair stands confidently.". It contains 1,221 images from the Naruto series and corresponding descriptive captions.

We use the same fine-tuning setup as for ArtBench-2.

## B.2 METRICS

### B.2.1 LINEAR DATAMODELING SCORE (LDS)

The Linear Datamodeling Score (LDS) (Park et al., 2023) evaluates a data attribution method, $\tau$, by assessing its ability to predict how model outputs change when the model is trained on different subsets of data.

Consider a full training dataset $\mathcal{S}$ of size $N$, a model output function $f(x; \theta)$ (e.g., the model's loss or logits for an input $x$ given parameters $\theta$), and the data attribution method $\tau$ under evaluation. The LDS calculation proceeds as follows:

1. **Generate Subsets:** Create $M$ random subsets of the training data, denoted as $\{\mathcal{S}^m\}_{m=1}^M$. Each subset $\mathcal{S}^m \subset \mathcal{S}$ typically has a fixed size, often a fraction $\alpha \cdot N$ of the total dataset size (e.g., $\alpha = 0.5$ for 50% subsets).

2. **For each query sample $x_{\text{query}}$ and each subset $\mathcal{S}^m$:**

   (a) **Calculate Actual Model Output ($y_m^{\text{actual}}$):** Retrain a new model (or fine-tune the existing model architecture) using only the training samples in the current subset $\mathcal{S}^m$. Let the parameters of this retrained model be $\theta_m$. The actual model output for the query $x_{\text{query}}$ using this subset-trained model is then $y_m^{\text{actual}} = f(x_{\text{query}}, \theta_m)$.

   (b) **Calculate Attribution-based Predicted Output ($y_m^{\text{predicted}}$):** This prediction is derived using the attribution method $\tau$. The attribution scores, $\tau(x_{\text{query}}, \mathcal{S})^{(i)}$, represent the influence of each training sample $x^{(i)}$ from the original full dataset $\mathcal{S}$ on the query $x_{\text{query}}$ (as determined by $\tau$ using the original model trained on $\mathcal{S}$). The predicted output for the subset $\mathcal{S}^m$ is the sum of these attribution scores for all samples $x^{(i)}$ that are part of $\mathcal{S}^m$:

   $$y_m^{\text{predicted}} = g_\tau(x_{\text{query}}, \mathcal{S}^m, \mathcal{S}) := \sum_{x^{(i)} \in \mathcal{S}^m} \tau(x_{\text{query}}, \mathcal{S})^{(i)} \tag{8}$$

3. **Compute LDS:** The LDS for the attribution method $\tau$ with respect to the query $x_{\text{query}}$ is the Spearman rank correlation $\rho$ (Spearman, 1961) between the list of $M$ actual model outputs and the list of $M$ attribution-based predicted outputs:

$$\text{LDS}(\tau, x_{\text{query}}) := \rho\left(\{y_m^{\text{actual}} : m \in [M]\}, \{y_m^{\text{predicted}} : m \in [M]\}\right)$$
$$= \rho\left(\{f(x_{\text{query}}, \theta_m) : m \in [M]\}, \{g_\tau(x_{\text{query}}, \mathcal{S}^m, \mathcal{S}) : m \in [M]\}\right) \tag{9}$$

A higher LDS value (closer to 1) indicates that the attribution scores provided by $\tau$ are more consistent with the true impact of training data subsets on the model's behavior for $x_{\text{query}}$, suggesting a more accurate attribution method. Conversely, an LDS closer to 0 suggests poor correlation.

**Implementation Details.** To conduct the LDS evaluation, we set $\alpha = 0.5$, and sample $M = 100$ ($M = 64$ for diffusion) different random subsets of the training set $\mathcal{D}$, and train models from scratch on each one of these subsets. For diffusion models, we let the model output function $f(x; \theta)$ be the simple loss $\mathcal{L}_{\text{Simple}}(x; \theta) = \mathbb{E}_{\epsilon, t}\left[||\epsilon - \epsilon_\theta(\sqrt{\bar{\alpha}_t}x + \sqrt{1 - \bar{\alpha}_t}\epsilon, t)||_2^2\right]$ from DDPM (Ho et al., 2020). To approximate the expectation, we select 50 timesteps evenly spaced within the interval $[1, T = 1000]$; and at each selected timestep, we sample one Gaussian noise $\epsilon \sim \mathcal{N}(0, \mathbf{I})$. Finally, we average the LDS scores over all the samples of interest.

For each LDS evaluation for diffusion, we utilize 500 query samples generated by the original model using distinct random seeds, and the average is reported. The query images are generated using the following dataset-specific prompting strategies:

- **CIFAR-2:** Query samples are generated unconditionally, as the DDPM for this dataset is trained without text prompts.
- **ArtBench-2:** 500 query images are generated with the prompt "a post-impressionism painting" and another 500 with "a ukiyo-e painting".
- **SB-Pokemon:** 1000 query images are generated using combinations of subject, style, and background categories from Table 8 (e.g., "A risograph print of Eevee, in a misty marsh"). The train/validation split of these categories is not considered for the LDS query set generation.
- **Naruto:** 1000 query images are generated with 1000 different prompts randomly sampled from the dataset.

For image classification task, we randomly selected 1,000 samples from the training set (that are non-overlapped with the 50,000 samples used for pre-training) for learning weights, and 1,000 samples from the validation set as queries for LDS evaluation.

For language modeling task, we used the training split of WikiText-103 (Merity et al., 2016) for pre-training, which contains 232,585 sequences of length 512. We used the 487 text sequences from the validation split as queries for LDS evaluation and the 523 sample from the test split for learning weights.

### B.2.2 Mislabeled Data Detection via Self-Influence

Data attribution can be used for the practical task of identifying noisy or corrupted labels within a training set, helping experts prioritize their attention (Koh & Liang, 2017; Pruthi et al., 2020).

The core hypothesis is that an incorrectly labeled example, being an outlier with respect to its assigned class, will exert a strong, corrective influence on the model. This is measured by its **self-influence**, i.e., the influence of a training point on its own loss, calculated as $\tau(x_i, x_i)$. A high self-influence score suggests that the model finds the example surprising or difficult to fit, which is characteristic of a mislabeled sample.

The evaluation procedure is as follows:

1. **Simulate Noisy Labels:** To create a ground truth, a fraction of the training data (10% in our experiments) is intentionally mislabeled. A model is then trained on this corrupted dataset.

2. **Rank by Self-Influence:** The self-influence score is computed for a 5,000-sample subset in the training set. All samples are then ranked in descending order based on these scores. Each self-influence score is normalized by dividing the sum of the scores of the top-ranked training samples. We found using the top-10 samples for normalization works the best for the mislabeled data detection task on ResNet-18 (top-100 for ViT).

3. **Compute AUC:** To quantify the quality of the ranking, we treat the known mislabeled samples as the "positive" class and the correctly labeled samples as the "negative" class. The Area Under the ROC Curve (AUC) is then computed. A higher AUC indicates that the self-influence scores are more effective at separating the mislabeled samples from the correctly labeled ones, with an ideal method ranking all mislabeled samples at the top. An AUC of 0.5 corresponds to a random ranking.

### B.2.3 Tail-Patch Score

The Tail-Patch Score (Chang et al., 2024) is an influence metric designed to measure the direct, causal impact of influential examples, rather than just correlational similarity. It quantifies the *additive contribution* of top-ranked proponents by measuring how much they improve the model's performance on a specific query through incremental training.

The procedure for a given query sample is as follows:

1. **Identify Proponents:** Use the data attribution method $\tau$ to identify the top-$k$ most influential training samples (proponents) for the query. In our experiments, we use $k = 128$ because the batch size we used for pretraining GPT-2 is 128.

2. **Form a Batch:** These $k$ proponents are collected and treated as a single training mini-batch.

3. **Perform Incremental Training:** Starting from the final, converged model checkpoint, perform a single, additional training step (a "tail-patch" step) using this mini-batch of proponents, maintaining all original training hyperparameters.

4. **Measure Probability Increase:** Measure the change in the model's log-probability of predicting the correct target sequence for the query before and after the tail-patch step.

The final Tail-Patch Score is this change in log-probability, averaged over all queries in the evaluation set. A higher score indicates that the attribution method is more effective at identifying training examples that are genuinely helpful for improving the model's performance on specific examples.

### B.3 Selected Attribution Methods

Our experimental setup for the baseline methods aligns with (Zheng et al., 2023) and (Park et al., 2023). Specifically, for all considered attribution methods, we incorporate TRAK's scalability

optimization (Park et al., 2023), which utilizes random projection of gradient features to enhance computational efficiency and reduce storage demands. Furthermore, also consistent with (Zheng et al., 2023), we exclusively use the final checkpoint of the trained model for all evaluations, omitting intermediate checkpoints to manage computational costs.

Next, we provide the detailed definitions of the baseline methods that we consider.

**TracIn.** We utilize the TracIn estimator as described by Pruthi et al. (2020). The attribution score for a training sample $x^n \in \mathcal{D}$ with respect to a query sample $x_{\text{query}}$ is implemented as:

$$\tau(x_{\text{query}}, x^n) = \frac{1}{C} \sum_{c=1}^{C} \left( P_c^\top \nabla_\theta \mathcal{L}_{\text{Simple}}(x_{\text{query}}; \theta^c) \right)^\top \cdot \left( P_c^\top \nabla_\theta \mathcal{L}_{\text{Simple}}(x^n; \theta^c) \right), \qquad (10)$$

where $C$ is the number of model checkpoints $\theta^c$ selected evenly from the training trajectory, $P_c$ is the projection matrix, and $\mathcal{L}_{\text{Simple}}(x; \theta^c)$ is the simple loss function from DDPM (Ho et al., 2020). Note that the TracIn method we report in our main paper is a specific instance of TracInCP where $C = 1$, meaning only the final model checkpoint is used. This aligns with our general approach for other baseline methods where we exclusively use the final checkpoint for computational efficiency, as detailed previously. For TracIn and other methods except Journey TRAK, each gradient is the average of gradients over all timesteps. We use the same projection dimension $k = 25472$ for all methods throughout our diffusion experiments. For image classification, $k = 11264$ for ResNet-18 and $k = 12800$ for ViT. For GPT-2 on language modeling tasks, TracIn and TRAK set $k = 196608$ and LoRA uses $k = 442,368$ with $r = 96$.

**TRAK.** We employ the TRAK estimator formulated by Park et al. (2023). For its application in our diffusion model setting, the model output function is instantiated as the simple loss in DDPM (Ho et al., 2020), $\mathcal{L}_{\text{Simple}}$. Consistent with our approach for other baselines, we use only the final model checkpoint, denoted as $\theta$. The TRAK attribution score for a training sample $x^n \in \mathcal{D}$ with respect to a query sample $x_{\text{query}}$ is then calculated as:

$$\tau_{\text{TRAK}}(x_{\text{query}}, x^n) = \left( P^\top \nabla_\theta \mathcal{L}_{\text{Simple}}(x_{\text{query}}; \theta) \right)^\top \left( \mathbf{\Phi}^\top \mathbf{\Phi} + \lambda I \right)^{-1} P^\top \nabla_\theta \mathcal{L}_{\text{Simple}}(x^n; \theta), \qquad (11)$$

$$\text{where } \mathbf{\Phi} = [\phi(x^{(1)}), \dots, \phi(x^{(N)})]^\top \text{ with } \phi(x^{(j)}) = P^\top \nabla_\theta \mathcal{L}_{\text{Simple}}(x^{(j)}; \theta). \qquad (12)$$

In these equations, $P$ represents the random projection matrix. The matrix $\mathbf{\Phi}$ is constructed from the projected gradients of all $N$ training samples $x^{(j)}$ in the dataset $\mathcal{D}$, and $\lambda$ is a regularization parameter for the kernel inversion for numerical stability. For each method except TracIn, we sweep the regularization parameter $\lambda$ over the values $[5 \times 10^{-3}, 5 \times 10^{-2}, 5 \times 10^{-1}, 5, 50]$.

**D-TRAK.** The implementation of D-TRAK is very similar to TRAK. The only difference is that the model output function is replaced by $\mathcal{L}_{\text{Square}}(x; \theta) = \mathbb{E}_{\epsilon,t} \left[ ||\epsilon_\theta(\sqrt{\bar{\alpha}_t}x + \sqrt{1 - \bar{\alpha}_t}\epsilon, t)||_2^2 \right]$.

$$\tau_{\text{D-TRAK}}(x_{\text{query}}, x^n) = \left( P^\top \nabla_\theta \mathcal{L}_{\text{Square}}(x_{\text{query}}; \theta) \right)^\top \left( \mathbf{\Phi}^\top \mathbf{\Phi} + \lambda I \right)^{-1} P^\top \nabla_\theta \mathcal{L}_{\text{Square}}(x^n; \theta), \qquad (13)$$

$$\text{where } \mathbf{\Phi} = [\phi(x^{(1)}), \dots, \phi(x^{(N)})]^\top \text{ with } \phi(x^{(j)}) = P^\top \nabla_\theta \mathcal{L}_{\text{Square}}(x^{(j)}; \theta). \qquad (14)$$

**Journey TRAK.** Journey TRAK (Georgiev et al., 2023) focuses on attributing influence to noisy images $x_t$ at a specific timestep $t$ throughout the generative process. In contrast, our approach aims to attribute the final generated image. Journey TRAK is adapted to our setting by averaging the attributions over the denoising timesteps as follows:

$$\tau_{\text{J-TRAK}}(x_{\text{query}}, x^n) = \frac{1}{T'} \sum_{t=1}^{T'} \left( P^\top \nabla_\theta \mathcal{L}_{\text{Simple}}^t(x_t; \theta) \right)^\top \left( \mathbf{\Phi}_{\text{TRAK}}^\top \mathbf{\Phi}_{\text{TRAK}} + \lambda I \right)^{-1} P^\top \nabla_\theta \mathcal{L}_{\text{Simple}}(x_t^n; \theta).$$

$$(15)$$

Here, $x_{\text{query}}$ is the final query image, and $x^n$ is the training sample from dataset $\mathcal{D}$. $T'$ denotes the total number of inference steps (set to 50 in our experiments), and $x_t$ is the noisy version of $x_{\text{query}}$ at inference step $t$ along the sampling trajectory. The matrix $\mathbf{\Phi}_{\text{TRAK}}$ is constructed identically to $\mathbf{\Phi}$ as defined in Equation equation 12 (i.e., using the average of projected gradients of $\mathcal{L}_{\text{Simple}}$ for all training samples with the final model parameters $\theta$). The term $\mathcal{L}_{\text{Simple}}^t(x_t; \theta)$ and $\mathcal{L}_{\text{Simple}}(x_t^n; \theta)$ are both evaluated for the noisy images at inference timestep $t$.

**DAS.** Diffusion Attribution Score (DAS) is a recently improved method for attributing influence to training samples in diffusion models (Lin et al., 2024). To attribute the influence of a training sample $x^n \in \mathcal{D}$ on a generated sample $x_{\text{query}}$ throughout the entire generation process, the attribution score $\tau_{\text{DAS}}(x_{\text{query}}, x^n)$ is calculated as follows:

$$\tau_{\text{DAS}}(x_{\text{query}}, x^n) = \| \frac{\left(P^\top \nabla_\theta \mathcal{L}_{\text{DAS}}(x_{\text{query}}; \theta)\right)^\top \left(\mathbf{\Phi}^\top \mathbf{\Phi} + \lambda I\right)^{-1} P^\top \nabla_\theta \mathcal{L}_{\text{DAS}}(x^n; \theta) \cdot r^n}{1 - \left(P^\top \nabla_\theta \mathcal{L}_{\text{DAS}}(x^n; \theta)\right)^\top \left(\mathbf{\Phi}^\top \mathbf{\Phi} + \lambda I\right)^{-1} P^\top \nabla_\theta \mathcal{L}_{\text{DAS}}(x^n; \theta)} \|^2, \quad (16)$$

$$\text{where } \mathbf{\Phi} = [\phi(x^{(1)}), \dots, \phi(x^{(N)})]^\top \text{ with } \phi(x^{(j)}) = P^\top \nabla_\theta \mathcal{L}_{\text{DAS}}(x^{(j)}; \theta). \quad (17)$$

$$\text{and } r^n = \frac{1}{T'} \sum_{t=1}^{T'} \|\epsilon_\theta(x_t^n, t) - \epsilon_t\|_2^2 \quad (18)$$

Here, $\mathcal{L}_{\text{DAS}}(x; \theta) = \mathbb{E}_{\epsilon, t} \left[\epsilon_\theta(\sqrt{\bar{\alpha}_t} x + \sqrt{1 - \bar{\alpha}_t} \epsilon, t)\right]$ and $r^n$ is the average of residual noise over all timesteps for the training sample $x^n$.

**LoGRA.** LoGRA (Low-Rank Gradient Approximation) is a method that significantly improves the efficiency of gradient projection for large models (Choe et al., 2024). Its key insight stems from the structure of backpropagation. For a layer with weights $W$, the gradient $\nabla W$ can be expressed as a sum of Kronecker products between the forward activations (layer inputs $x_i$) and backward activations (output gradients $Dx_o$):

$$\text{vec}(\nabla W) = \sum_{t=1}^{T} x_{i,t} \otimes Dx_{o,t}. \quad (19)$$

Instead of first computing the full, high-dimensional gradient $\nabla W$ and then projecting it, LoGRA applies separate, smaller projection matrices, $P_i$ and $P_o$, directly to the activations:

$$g_W(x) = \sum_{t=1}^{T} (P_i x_{i,t}) \otimes (P_o Dx_{o,t}), \quad (20)$$

where $g_W(x)$ is the projected gradient for the layer's weights. The final feature vector for the entire model, $\phi_{\text{LoGRA}}(x)$, is a concatenation of these efficiently computed projected gradients from all targeted layers. This "project-then-combine" strategy avoids ever materializing the full model gradients, leading to substantial memory and computational savings. The final attribution score follows the TRAK formulation (Eq. 11) using these features.

**EKFAC.** The Eigenvalue-corrected K-FAC (EKFAC) method scales influence functions by constructing a high-quality, tractable approximation of the inverse Hessian matrix (Grosse et al., 2023). For our EKFAC baseline, we adopt a hybrid approach: we compute gradient features using the efficient LoGRA projection (Eq. 20), but replace the standard empirical kernel with the EKFAC Hessian approximation.

The EKFAC kernel, $\mathcal{H}_{\text{EKFAC}}^{-1}$, is built by approximating the full Hessian $\mathcal{H}$ as a block-diagonal matrix, with one block $\mathcal{H}_l$ for each model layer $l$. Each block is then approximated by the Kronecker product of two smaller matrices:

$$\mathcal{H} \approx \text{diag}(\mathcal{H}_1, \dots, \mathcal{H}_L), \quad \text{where} \quad \mathcal{H}_l \approx A_l \otimes G_l. \quad (21)$$

Here, $A_l$ is the covariance of layer inputs and $G_l$ is the covariance of layer output gradients. The inverse is then computed efficiently with eigenvalue corrections for stability, as detailed in the original work. The final attribution score is calculated as:

$$\tau_{\text{EKFAC}}(x_{\text{query}}, x^n) = \phi_{\text{LoGRA}}(x_{\text{query}})^\top \cdot \mathcal{H}_{\text{EKFAC}}^{-1} \cdot \phi_{\text{LoGRA}}(x^n). \quad (22)$$

This hybrid approach leverages LoGRA's efficiency for feature generation and EKFAC's sophisticated Hessian approximation for the kernel.

### B.4 WEIGHT LEARNING

The self-supervised weight learning procedure employed in our experiments is detailed in Algorithm 1. For general-purpose weight learning, the query set consists of $N_q = 500$ samples, generated using

---

**Algorithm 1** Parameter Group Weight Learning

---

**Require:** Set of $N_q$ query samples $\{x_q^{(i)}\}_{i=1}^{N_q}$.

**Require:** For each query $x_q^{(i)}$: precomputed group-wise contributions $C^{(i)}$, an $N \times M$ matrix where $C^{(i)}[n, j]$ is the $j$-th group's attribution score of the $n$-th training sample on query $x_q^{(i)}$. ($N$: number of training samples, $M$: number of parameter groups).

**Require:** Hyperparameters: learning rate $\eta$, number of epochs $E_{\max}$, the number of top-ranked samples to be considered $k$, weight regularization $\lambda'$.

**Ensure:** Learned parameter group weights $w \in \mathbb{R}^M$.

1: Initialize raw weights $w_{\mathrm{raw}} \in \mathbb{R}^M$ with random values.
2: Initialize optimizer $\mathcal{O}$ (e.g., AdamW) for $w_{\mathrm{raw}}$ with learning rate $\eta$ and weight decay $\lambda'$.
3: **for** epoch $e = 1$ to $E_{\max}$ **do**
4:     **for** each query $x_q^{(i)}$ **do**
5:         Let $C_q \leftarrow C^{(i)}$ (the $N \times M$ matrix for current query $x_q^{(i)}$).
6:         $w \leftarrow \mathrm{Softmax}(w_{\mathrm{raw}})$                  $\triangleright$ Derive actual weights $w_j$ from $w_{\mathrm{raw},j}$
7:         Calculate attribution scores $S \in \mathbb{R}^N$ for query $x_q^{(i)}$:
8:         $S_n \leftarrow \sum_{j=1}^M w_j \cdot C_q[n, j]$ for $n = 1, \ldots, N$.
9:         $S \leftarrow S/\|S\|_2$                        $\triangleright$ Normalize scores
10:        $S_{\mathrm{ranked}} \leftarrow$ reordered scores from sorting $S$ in descending order.
11:        $\mathcal{L}_{\mathrm{SSL}} \leftarrow -\mathrm{Avg}(S_{\mathrm{ranked}}[1 \ldots k])$      $\triangleright$ Self-supervised loss (Eq. 6)
12:        $\mathcal{O}.\mathrm{zero\_grad}()$
13:        $\mathrm{loss.backward}()$
14:        $\mathcal{O}.\mathrm{step}()$                      $\triangleright$ Update $w_{\mathrm{raw}}$
15:     **end for**
16: **end for**
17: Obtain final weights $w = \mathrm{Softmax}(w_{\mathrm{raw}})$ .
18: **return** $w$

---

random seeds and diverse prompts from the respective dataset. For fine-grained weight learning (e.g., to discern influences of specific semeantic elements like backgrounds), a more targeted set of $N_q = 200$ samples is utilized, generated with prompts that emphasize a particular semantic element (e.g., for background, "*in a misty marsh*" , "*on a rocky beach*", while omitting explicit subject or style specifications). Standard training settings include $E_{\max} = 10$ epochs and a learning rate $\eta = 0.01$, managed by an AdamW optimizer with a cosine annealing schedule. For each weight learning task, we conduct a hyperparameter search, varying $k_{\mathrm{top}}$ (the number of top- and bottom-ranked samples for the contrastive loss) across the values $[1, 5, 10, 20, 50, 100, 200, 500, 1000, 5000]$, and the weight regularization parameter $\lambda'$ over $[0.0, 0.02, 0.1, 0.2, 0.3, 0.4, 0.5, 0.8, 1.0, 1.5]$. The primary computational cost lies in generating the query samples and pre-computing their corresponding group-wise contributions $C^{(i)}$, which scales with $N_q$, the cost of the generative model, and the chosen attribution method. Once these prerequisites are precomputed, the weight optimization process itself is efficient, typically concluding in a few seconds.

## C  ADDITIONAL RESULTS

### C.1  CONSISTENCY OF PER-GROUP ATTRIBUTION SIGNALS

In Section 3, we demonstrated that attribution strength is highly heterogeneous across a model's parameters. A critical question is whether this pattern of heterogeneity is a stable, intrinsic property of the model, or an artifact of a specific dataset or attribution method. To validate the former, we conduct a similarity analysis.

For a given model, we can characterize its pattern of heterogeneity by a vector containing the per-group LDS scores. We compute these vectors for multiple settings, varying both the dataset (ArtBench-2, Naruto, SB-Pokemon) and the base attribution method (D-TRAK, TracIn), and then measure the cosine similarity between them. A high similarity indicates that the underlying pattern of which parameter groups are most important is consistent across different settings.

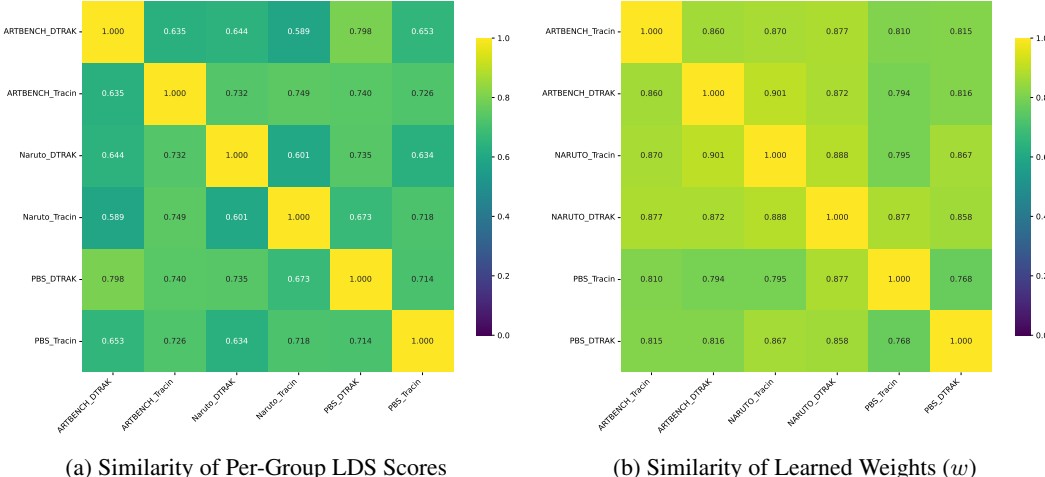

(a) Similarity of Per-Group LDS Scores

(b) Similarity of Learned Weights ($w$)

Figure 6: **Consistency of Attribution Signals and Learned Weights.** Heatmaps show the cosine similarity between (a) vectors of raw per-group LDS scores and (b) our learned weight vectors, across different datasets and methods. Brighter colors indicate higher similarity. Our learned weights show a significantly more stable and robust pattern.

Figure 6a visualizes the similarity matrix for these per-group LDS scores. The heatmap shows a consistent, moderately strong correlation across all pairs (average off-diagonal similarity of 0.689). Given that individual per-group LDS measurements are inherently noisy, this level of similarity provides strong evidence that a stable, underlying pattern of parameter importance exists within the model architecture.

Furthermore, we hypothesized that if our method is effective, it should be able to filter the noise in these raw scores and distill a more robust representation of this underlying pattern. To test this, we performed the same similarity analysis on the *learned parameter weights* ($w$) from each setting. The result, shown in Figure 6b, confirms this hypothesis. The learned weights are significantly more consistent, with a much higher average off-diagonal similarity of 0.845. This is a powerful validation of our approach: it demonstrates that our self-supervised learning algorithm successfully identifies and amplifies the stable, intrinsic signal of layer-wise importance while filtering out the noise present in the raw per-group attribution scores.

### C.2    SENSITIVITY ANALYSIS FOR HYPERPARAMETER $k$

Our self-supervised learning objective (Eq. 6) depends on the hyperparameter $k$, which defines the number of top-scoring training examples to use as pseudo-positives. To assess the stability and sensitivity of our method to this choice, we performed a sensitivity analysis where we varied $k$ across a range of values: $\{1, 5, 10, 20, 50, 100\}$.

Table 9 presents the results, showing the *improvement* in LDS ($\Delta$LDS) achieved by our method over the unweighted baseline for different values of $k$. The analysis was conducted for both TracIn and TRAK on the ArtBench-2 and SB-Pokemon datasets.

The results reveal two key findings. First, on datasets like ArtBench-2, the performance is remarkably stable across a broad range of $k$, indicating that the learning process is robust and not dependent on fine-grained tuning.

Second, on datasets like SB-Pokemon, performance is more sensitive and can degrade for larger values of $k$. This is expected behavior: as $k$ increases, the set of pseudo-positives is more likely to include noisy, non-influential examples, which can corrupt the supervisory signal for weight learning.

Despite this sensitivity, the method remains highly practical for two reasons. First, the weight-learning process is extremely efficient, typically converging in under one minute, which makes tuning $k$ for a given dataset or task entirely feasible. Second, as we show in Appendix C.5, the learned weights often

Table 9: **Sensitivity to** $k$**.** The table shows the final LDS (%) for different values of $k$, the number of pseudo-positives. The "w/o $w$" column is the unweighted baseline, while the "$w$" column shows the result using the best-tuned $k$ as reported in the main text. Performance is stable for a range of smaller $k$ but can degrade if $k$ becomes too large.

| Method | Dataset | w/o $w$ | k=1 | k=5 | k=10 | k=20 | k=50 | k=100 |
|--------|---------|---------|------|------|------|------|------|-------|
| TracIn | ArtBench-2 | 17.63 | 21.07 | 22.01 | **22.02** | 21.97 | 21.91 | 21.92 |
| TRAK | ArtBench-2 | 18.39 | 21.75 | 21.92 | **22.15** | 22.07 | 21.97 | 21.89 |
| TracIn | SB-Pokemon | 9.34 | 11.63 | **11.79** | 11.73 | 7.46 | 8.78 | 6.23 |
| TRAK | SB-Pokemon | 10.68 | 11.83 | **12.24** | 11.97 | 11.00 | 8.98 | 9.00 |

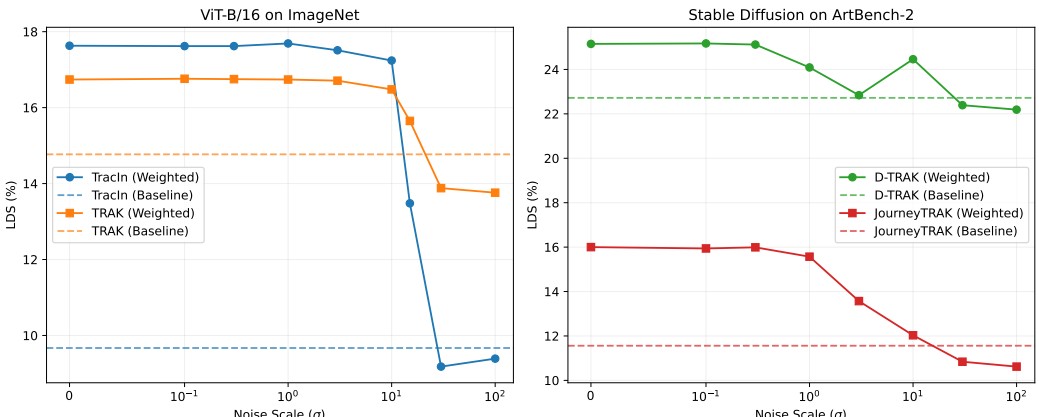

Figure 7: **Robustness to Noise during Weight Learning.** LDS obtained after learning weights from attribution scores perturbed with Gaussian noise of standard deviation $\sigma$. The weighted methods maintain their advantage over a broad range of noise levels, degrading only when the injected noise completely dominates the original signal.

demonstrate strong transferability across different settings, reducing the need for repeated tuning for the same model.

### C.3 SENSITIVITY ANALYSIS FOR NOISE IN ATTRIBUTION SCORES

To investigate if our weight-learning algorithm is robust to noise in the attribution scores, we perturb the layer-wise attribution scores during weight learning. Concretely, we first measure the standard deviation $\sigma_j$ of the attribution scores for each parameter group $j$ across all query-training pairs. Then, we add i.i.d. Gaussian noise $\epsilon_j \sim \mathcal{N}(0, (s \cdot \sigma_j)^2)$ to each contribution $a_j(x_q, x_n)$ before computing the top-$k$ objective, where $s$ is the noise scale multiplier. Finally, we evaluate the learned weights on clean validation queries. Figure 7 reports the resulting LDS as a function of the noise scale multiplier $s$ for both ViT-B/16 on ImageNet (TracIn and TRAK) and Stable Diffusion on ArtBench-2 (D-TRAK and JourneyTRAK).

Across all settings, the weighted methods retain a sizeable lead over their unweighted baselines even when $\sigma$ is one to two orders of magnitude larger than the typical score magnitude. For ViT, the TracIn-based weights remain virtually unchanged up to $\sigma = 10$, and only begin to converge towards the unweighted model beyond $\sigma = 30$. The TRAK variant exhibits the same trend, degrading smoothly rather than collapsing. The diffusion experiments mirror this behavior: D-TRAK maintains its LDS improvement for small and moderate noise, while JourneyTRAK stays above its baseline up to $\sigma = 10$. These results corroborate the intuition discussed in the rebuttal that our objective only requires the top-$k$ set to be enriched with true influences on average; sporadic corruptions behave like zero-mean noise that is averaged out during weight learning.

Table 10: **LDS (%) evaluation of alternative loss functions on ArtBench-2.** Our proposed self-supervised top-$k$ loss consistently achieves the best or near-best improvements. We bold the highest-performing loss for each attribution method. The 'Top-k' column corresponds to our main proposed loss (Eq. 6).

| Method | w/o $w$ | Top-k (Eq. 6) | Supervised | $-$ Bot-k | Top-k $-$ Bot-k | No Norm |
|--------|---------|---------------|------------|-----------|-----------------|---------|
| TracIn | 17.63 $\pm 0.96$ | 22.02 $\pm 1.05$ | **22.31** $\pm 0.99$ | 21.28 $\pm 1.06$ | 21.66 $\pm 1.06$ | 6.89 $\pm 0.81$ |
| TRAK | 18.39 $\pm 0.96$ | **22.15** $\pm 0.99$ | 21.83 $\pm 0.99$ | 20.95 $\pm 1.00$ | 21.83 $\pm 1.00$ | 9.80 $\pm 0.85$ |
| JTRAK | 11.56 $\pm 0.89$ | **16.00** $\pm 1.00$ | 13.87 $\pm 0.94$ | 13.96 $\pm 0.96$ | 15.50 $\pm 1.05$ | 1.62 $\pm 0.80$ |
| DTRAK | 22.72 $\pm 1.02$ | **25.15** $\pm 1.09$ | 24.50 $\pm 1.06$ | 20.37 $\pm 0.94$ | 24.00 $\pm 1.05$ | 14.69 $\pm 0.85$ |
| DAS | 30.47 $\pm 1.15$ | **31.58** $\pm 1.15$ | 30.90 $\pm 1.15$ | 31.03 $\pm 1.16$ | **31.58** $\pm 1.14$ | 7.72 $\pm 1.41$ |

## C.4 ABLATION STUDY OF THE LOSS FUNCTION

In the main text, we propose a self-supervised loss function (Eq. 6) to learn the parameter weights $w$. To validate this choice, we experimented with several alternative formulations. For each alternative, we performed a hyperparameter sweep to ensure a fair comparison. The results, evaluated on the ArtBench-2 dataset, are presented in Table 10. Below, we detail each of the alternative objectives.

**Supervised Loss with Data Augmentation.** A natural alternative is a supervised approach. We generate a query sample $x'_{\text{query}}$ by applying data augmentations (random horizontal flip, resized crop, color jitter) to a training sample $x^n$. We then treat $x^n$ as the ground-truth "pseudo-positive" for the query $x'_{\text{query}}$. The loss function is formulated to maximize the score of this single pseudo-positive, while keeping the normalization term:

$$\mathcal{L}_{\text{Supervised}}(w) = -\frac{\widetilde{\tau}(x'_{\text{query}}, x^n; w)}{\|\widetilde{\boldsymbol{\tau}}(x'_{\text{query}}, \mathcal{D}; w)\|_2}.$$

As shown in Table 10, this approach is competitive but generally outperformed by our self-supervised loss. While it avoids the hyperparameter $k$, it introduces a new dependency on the choice of data augmentations, which can be highly domain-specific and may not always preserve the core semantic content needed for attribution.

**Bottom-k Loss.** Another potential source of signal lies in the least influential (most negative) examples. We test a loss function that minimizes the average score of the bottom-$k$ examples:

$$\mathcal{L}_{\text{Bot-k}}(w) = \frac{\frac{1}{k}\sum_{j\in\mathcal{I}_{\text{bot-k}}}\widetilde{\tau}_j}{\|\widetilde{\boldsymbol{\tau}}(w)\|_2}.$$

In theory, if negative influence signals are strong, minimizing them should improve attribution. However, as shown in Table 10, this objective yields mixed results: while it often improves over the unweighted baseline, it consistently underperforms the Top-$k$ objective and sometimes even degrades performance (D-TRAK). We reasonably hypothesize that truly harmful training examples are rare, and the bottom tail of the attribution score distribution is largely dominated by noise rather than meaningful negative signal; neural networks tend to "memorize' specific examples but does not actively "forget' the exact opposite of a point.

**Top-k minus Bottom-k Loss.** A common strategy in contrastive learning is to maximize the gap between positive and negative examples. We implemented this by subtracting the average score of the bottom-$k$ examples from the numerator:

$$\mathcal{L}_{\text{Gap}}(w) = -\frac{\frac{1}{k}\sum_{i\in\mathcal{I}_{\text{top-k}}}\widetilde{\tau}_i - \frac{1}{k}\sum_{j\in\mathcal{I}_{\text{bot-k}}}\widetilde{\tau}_j}{\|\widetilde{\boldsymbol{\tau}}(w)\|_2}.$$

Intuitively, this combines the Top-$k$ signal with the Bottom-$k$ minimization. However, we found that this did not provide a consistent benefit and sometimes slightly underperformed our main loss. This further suggests that the bottom-$k$ scores are significantly dominated by noise ($\epsilon_j$), so including them adds variance while providing little useful signal. Thus, focusing on the top-$k$ positive signal (as in our main objective) remains the most robust strategy.

Table 11: **Generalization of Learned Weights across Datasets.** We report LDS (%) for: the standard approach ("w/o $w$"); with parameter weights learned on the respective dataset ("$w$"); and with parameter weights learned on the *other* dataset shown ("$w_{\text{other}}$", e.g., for ArtBench-2, $w_{\text{other}}$ uses weights from Naruto, and vice-versa). Higher LDS values indicate better performance. TRAK's random projection was applied to all methods and datasets for fair comparison.

| Method | ArtBench-2 | | | Naruto | | |
|--------|------------|------|------------------|--------|------|------------------|
| | w/o $w$ | $w$ | $w_{\text{other}}$ | w/o $w$ | $w$ | $w_{\text{other}}$ |
| TracIn | 17.63 | **22.02** | 20.19 | 10.54 | 13.59 | **13.48** |
| TRAK | 18.39 | **22.15** | 19.35 | 14.61 | 17.02 | **17.19** |
| JTRAK | 11.56 | **16.00** | 12.55 | 13.41 | **14.56** | 14.36 |
| DTRAK | 22.72 | **25.15** | 23.61 | 16.75 | 17.85 | **18.36** |
| DAS | 30.47 | **31.58** | 30.74 | 18.72 | 20.44 | **20.45** |

**No Normalization.** To validate the role of the denominator in our loss, we performed an ablation where we removed the $\ell_2$ norm, leaving only the numerator to be maximized. As the "No norm" column in Table 10 shows, this completely destabilizes the learning process, leading to a drastic drop in performance. This result strongly confirms the importance of the normalization term, which, as argued in our SNR derivation, acts as a crucial regularizer that estimates the noise level. Without it, the optimization can trivially maximize the numerator by simply scaling up the weights $w$, failing to learn a meaningful signal of parameter importance.

These experiments validate that our proposed self-supervised loss provides the most robust and effective performance across a range of attribution methods, offering a well-balanced objective that is both theoretically grounded and empirically superior to these alternatives.

## C.5 Generalization of Learned Weights

Our experiments demonstrate that the learned parameter importance weights exhibit notable generalization capabilities, not only across different datasets but also across various attribution methods. This suggests a degree of robustness and potential for transferring learned weights, potentially mitigating the need for re-learning them in every new scenario. In most cases, applying weights learned from a different context (either another dataset or another attribution method) still results in an improvement in the Linear Datamodeling Score (LDS) compared to not using weights.

**Generalization Across Datasets.** As shown in Table 11, weights learned on one dataset often yield performance improvements when applied to a different dataset. For instance, when evaluating on Naruto, applying weights learned from ArtBench-2 consistently leads to higher LDS scores across all attribution methods compared to using weights learned directly on Naruto. Conversely, when evaluating on ArtBench-2, weights learned on ArtBench-2 itself generally provide the best results. This observation suggests that the characteristics of the dataset used for weight learning can influence the quality and transferability of the learned weights. It is plausible that a larger or more diverse dataset, such as ArtBench-2 in our experiments, might facilitate the learning of more universally effective weights. However, the precise factors contributing to this phenomenon remain an avenue for future investigation.

**Generalization Across Attribution Methods.** Table 12 details the performance when weights learned using one attribution method are applied to another, evaluated on the ArtBench-2 dataset. A key observation is that for most methods (TracIn, TRAK, D-TRAK, DAS), the highest LDS is achieved when using weights learned for the method itself (i.e., the values in the "$w$" column). Nevertheless, applying weights learned by other methods often leads to LDS improvements over the baseline "w/o $w$" scenario.

Overall, these findings underscore the adaptability of the learned weights. While dataset-specific and method-specific weights often yield optimal results, the ability of these weights to generalize offers practical advantages, particularly in scenarios where re-computing weights for every specific dataset-method combination might be computationally prohibitive.

Table 12: **Generalization of Learned Weights across Methods.** We report LDS (%) for each method on ArtBench-2: without any learned weights ("w/o $w$"); with weights learned by the method itself ("$w$"); and with weights learned by other attribution methods (e.g., "$w_{\text{TracIn}}$" signifies that weights learned by TracIn were applied to the method in the current row). Higher LDS values indicate better performance. TRAK's random projection was applied to all methods.

| Method | ArtBench-2 | | | | | | |
|---|---|---|---|---|---|---|---|
| | w/o $w$ | $w$ | $w_{\text{TracIn}}$ | $w_{\text{TRAK}}$ | $w_{\text{JTRAK}}$ | $w_{\text{DTRAK}}$ | $w_{\text{DAS}}$ |
| TracIn | 17.63 | **22.02** | —— | 21.02 | 20.28 | 19.96 | 19.97 |
| TRAK | 18.39 | **22.15** | 21.80 | —— | 20.63 | 19.68 | 20.82 |
| JourneyTRAK | 11.56 | **16.00** | 15.64 | 15.58 | —— | 12.16 | 14.05 |
| D-TRAK | 22.72 | **25.15** | 23.60 | 23.45 | 22.07 | —— | 23.91 |
| DAS | 30.47 | **31.58** | 30.14 | 30.71 | 28.69 | 29.68 | —— |

Table 13: **Tail-patch score (%) on OpenWebText with Llama3-8B-Instruct.** $\Delta$ denotes the average change per query with 95% CI half-width. Higher is better.

| Method | w/o $w$ | $w$ | $(\Delta)$ |
|---|---|---|---|
| Random | -0.07 | — | — |
| TracIn | 0.04 | 0.65 | 0.61 $_{\pm 0.11}$ |
| LoGRA | 1.78 | 1.92 | 0.15 $_{\pm 0.05}$ |

## C.6 SCALABILITY TO LARGER MODELS

To address the concern regarding the scalability of our method, we conduct an additional experiment using Llama3-8B-Instruct (Grattafiori et al., 2024). Following the protocol in Choe et al. (2024), we perform data attribution on the OpenWebText (OWT) dataset (Gokaslan et al., 2019), assuming that the model was pre-trained on this corpus, without conducting further training ourselves. We extract the first 1 million entries from the dataset and process them into 99,642 sequences of fixed length 512 tokens to form the training set. We then generate 500 queries, each consisting of 512 tokens, by prompting the model to continue text from 500 snippets randomly sampled from the training set. We employ the tail-patch evaluation: for each query, we take the top-1 attributed training example, perform one additional fine-tuning step of the model on that example, and measure the resulting change in the query's log-likelihood. The results, reported in Table 13, show that the weighted attribution consistently outperforms the unweighted baselines, mirroring the patterns observed for smaller models in the main text.

## C.7 QUALITATIVE ANALYSIS FOR LANGUAGE MODELING

To provide more insights into the qualitative effectiveness of our method, we provide examples of data attribution results, comparing the top-1 ranked training example retrieved by the baseline attribution method and the weighted counterpart. The results are shown in Figures 8, 9 for GPT-2-small and in Figure 10 for Llama3-8B-Instruct.

## D USAGE OF LARGE LANGUAGE MODELS

Following ICLR 2026 policy, hereby we acknowledge that we used LLMs for proofreading and polishing the main text of this paper, and structuring our draft into formal sections B.2.2, B.2.3, and C.4 in the Appendix.

---

**Query**

August 31, the same night that the NK I Corps offensive rolled across the river.

Near the end of the month two reconnaissance patrols from the 9th Infantry had crossed to the west side of the Naktong and observed North Korean tank and troop activity 2 miles (3.2 km) west of the river. Information obtained later indicated it was in fact the command post of the NK 9th Division. On August 25, 9th Infantry commander Colonel John G. Hill outlined projected " Operation Manchu, " which was to be a company-sized combat patrol to cross the river, advance to the suspected North Korean command post and communications center, destroy it, capture prisoners, and collect intelligence.

The 9th Infantry Regiment had planned Task Force Manchu on orders from the 2nd Division commander Major General Laurence B. Keiser, which in turn had received instructions from Eighth United States Army commander Lieutenant General Walton Walker for aggressive patrolling. Keiser decided the patrol should cross the river at the Paekchin ferry. The 9th Infantry reserve, E Company, reinforced with one section of light machine guns from H Company, was to be the attack force. The 1st Platoon, 2nd Engineer Combat Battalion, was to transport it across the river in assault boats the night of August 31. Two heavy weapons companies, D and H, were each to furnish one section of heavy machine guns, one section of 81-mm. mortars, and one section of 75-mm. recoilless rifles for supporting fires. A platoon of 4.2-inch mortars was also to give support.

After dark on August 31, First Lieutenant Charles I. Caldwell of D Company and First Lieutenant Edward Schmitt of H Company, 9th Infantry, moved their men and weapons to the base of Hill 209, which was within B Company 's defense sector and overlooked the Paekchin ferry crossing of the Naktong River. The raiding force, E Company, was still in its regimental reserve position about 2 miles (3.2 km) west of Yongsan, getting ready with the engineer platoon to move to the crossing site. Colonel Hill went forward in the evening with the 4.2-inch mortar platoon to its position at the base of Hill 209 where the mortarmen prepared to set up their weapons.

By 21:00, the closest front line unit was B Company on top of

---

**Top-1 Retrieved by Baseline EKFAC**

= = Composition = =

Glen Boyd of Blogcritics said that the album has " Deep-ass bass lines, old-school funk samples, and plenty of street smart ghetto attitude are what powers this record. " Jerry Heller wrote that Eazy raps more up front on the album than he does on Straight Outta Compton, and insists that the album 's lyrics contain more sexual humor than gangsta vibe.

The album 's title track and lead single " Eazy-Duz-It ", written by MC Ren, opens with a woman acclaiming Eazy-E 's style. Eazy then interrupts saying " Bitch shut the fuck up, get the fuck outta here. " This is followed by a bass line provided by Dr. Dre. Soon, Eazy begins to rap about himself and things that he does. The song declares that Eazy is a " hardcore villain " who collects money from his prostitutes, and feels great when his " pockets are fat. " The chorus, repeated three times, states that he " is a gangsta having fun ". The piece is laden with the aural mainstays of gangsta rap, including gunshots, and references to several drugs.

" Boyz n the Hood " was written by Ice Cube, with some contribution by Eazy-E. The song is about growing up in Compton, California, and describes the gangster lifestyle. It conceives the " ghetto landscape as a generalized abstract construct … [ and ] also introduces a localized nuance that conveys a certain proximity, effectively capturing a narrowed sense of place through which young thugs and their potential crime victims move in tandem, " as put by cultural historian Murray Forman.

" No More ? ' s " is similar to " Boyz n the Hood " in its theme. The piece begins with an interview between Eazy and a female journalist, who asks about his childhood. Eazy explains (in verse) that he was ruthless, in a gang, " specialized in gankin, " (loosely, to steal from) and had no respect for rules. He is then asked if he has ever been in an armed robbery. He responds, " You mean a 211 ? " The following verses tell of Eazy 's exploits as a thief and thug.

= = Reception = =

= = = Commercial performance = = =

The

---

**Top-1 Retrieved by Weighted EKFAC (Ours)**

1935 Eichmann married Veronika (Vera) Liebl (1909 – 93). The couple had four sons: Klaus (b. 1936 in Berlin), Horst Adolf (b. 1940 in Vienna), Dieter Helmut (b. 1942 in Prague) and Ricardo Francisco (b. 1955 in Buenos Aires). Eichmann was promoted to SS-Hauptscharführer (head squad leader) in 1936 and was commissioned as an SS-Untersturmführer (second lieutenant) the following year.

Nazi Germany used violence and economic pressure to encourage Jews to leave Germany of their own volition; around 250,000 of the country 's 437,000 Jews emigrated between 1933 and 1939. Eichmann travelled to British Mandatory Palestine with his superior Herbert Hagen in 1937 to assess the possibility of Germany 's Jews voluntarily emigrating to that country, disembarking with forged press credentials at Haifa, whence they travelled to Cairo in Egypt. There they met Feival Polkes, an agent of the Haganah, with whom they were unable to strike a deal. Polkes suggested that more Jews should be allowed to leave under the terms of the Haavara Agreement, but Hagen refused, surmising that a strong Jewish presence in Palestine might lead to their founding an independent state, which would run contrary to Reich policy. Eichmann and Hagen attempted to return to Palestine a few days later, but were denied entry after the British authorities refused them the required visas. They prepared a report on their visit, which was published in 1982.

In 1938, Eichmann was posted to Vienna to help organise Jewish emigration from Austria, which had just been integrated into the Reich through the Anschluss. Jewish community organisations were placed under supervision of the SD and tasked with encouraging and facilitating Jewish emigration. Funding came from money seized from other Jewish people and organisations, as well as donations from overseas, which were placed under SD control. Eichmann was promoted to SS-Obersturmführer (first lieutenant) in July 1938, and appointed to the Central Agency for Jewish Emigration in Vienna, created in August. By the time he left Vienna in May 1939, nearly 100,000 Jews had left Austria legally, and many more had been smuggled out to Palestine and elsewhere.

= = Second World War = =

---

Figure 8: **Qualitative Analysis for GPT-2-small.** Comparison of the most influential training example retrieved by the baseline EKFAC and our weighted EKFAC. The weighted method retrieves a training example that is semantically more relevant to the query.

**Query**

in the eastern provinces of Rome and with the Parthians as well. The peace between Parthia and Rome lasted 50 years until Emperor Trajan of Rome invaded Armenia in 114.
= = = Other major power struggles and rebellions = = =
The war with Parthia was not Nero 's only major war but he was both criticized and praised for an aversion to battle. Like many emperors, Nero faced a number of rebellions and power struggles within the empire.
British Revolt of 60 – 61 (Boudica 's Uprising)
In 60, a major rebellion broke out in the province of Britannia. While the governor Gaius Suetonius Paulinus and his troops were busy capturing the island of Mona (Anglesey) from the druids, the tribes of the southeast staged a revolt led by queen Boudica of the Iceni. Boudica and her troops destroyed three cities before the army of Paulinus could return, receive reinforcements, and quell the rebellion in 61. Fearing Paulinus himself would provoke further rebellion, Nero replaced him with the more passive Publius Petronius Turpilianus.
The Pisonian Conspiracy of 65
In 65, Gaius Calpurnius Piso, a Roman statesman, organized a conspiracy against Nero with the help of Subrius Flavus and Sulpicius Asper, a tribune and a centurion of the Praetorian Guard. According to Tacitus, many conspirators wished to " rescue the state " from the emperor and restore the Republic. The freedman Milichus discovered the conspiracy and reported it to Nero 's secretary, Epaphroditos. As a result, the conspiracy failed and its members were executed including Lucan, the poet. Nero 's previous advisor, Seneca was ordered to commit suicide after admitting he discussed the plot with the conspirators.
The First Jewish War of 66 – 70
In 66, there was a Jewish revolt in Judea stemming from Greek and Jewish religious tension. In 67, Nero dispatched Vespasian to restore order. This revolt was eventually put down in 70, after Nero 's death. This revolt is famous for Romans breaching the walls of Jerusalem and destroying the Second Temple of Jerusalem.
= = = The revolt of Vindex and Galba and the death of Nero = = =
In March 68, Gaius

**Top-1 Retrieved by Baseline EKFAC**

, Leslie Fuller, Nikki Webber, and Terrence McDonnell. For its first two seasons the syndicated version had Deirdre Cossman for its managing producer, then Dennis F. McMahon became producer for the next two seasons (joined by Dominique Bruballa as his line producer), after which Jennifer Weeks produced the next four seasons of syndicated Millionaire shows, initially accompanied by Amanda Zucker as her line producer, but later joined for the 2008 – 09 season by Tommy Cody (who became sole producer in the 2009 – 10 season). The first 65 shuffle format episodes were produced by McPaul Smith, and as of 2011, the title of producer is held by Bryan Lasseter. The network version had Ann Miller and Tiffany Trigg for its supervising producers; they were joined by Wendy Roth in the first two seasons, and by Michael Binkow in the third and final season. After Rubino 's promotion to co-executive producer, the syndicated version 's later supervising producers included Sirop (2004 – 09), Geena Gintzig (2009 – 10), Brent Burnette (2010 – 12), Geoff Rosen (2012 – 14), and Liz Harris (2014 – 16).
The original network version of Millionaire was directed by Mark Gentile, who later served as the syndicated version 's consulting producer for its first two seasons, and then as the director of Duel, which ran on ABC from December 2007 to July 2008. The syndicated version was directed by Matthew Cohen from 2002 to 2010, by Rob George from 2010 to 2013, and by Brian McAloon in the 2013 – 14 season. Former Price Is Right director Rich DiPirro became Millionaire 's director in 2014.
= = Production = =
The U.S. version of Millionaire is a co-production of 2waytraffic, a division of Sony Pictures Entertainment, and Valleycrest Productions, a division of The Walt Disney Company. 2waytraffic purchased Millionaire 's original production company Celador in 2008, while Valleycrest has produced the series since its beginning, and holds the copyright on all U.S. Millionaire episodes to date. The show is distributed by Valleycrest 's corporate sibling Disney – ABC Domestic Television (previously known as Buena Vista Television).
The U.S. Millionaire was taped at ABC 's Television Center East studio on the Upper West Side of Manhattan

**Top-1 Retrieved by Weighted EKFAC (Ours)**

house arrest, he signed a plea bargain for minor offenses, connected to bringing in foreign currency without a permit, and was sentenced to community service and judicial supervision for three years. In view of the Abergil brothers extradition to the US, he and his wife were arrested for violating immigration laws. It is believed that American authorities tried to apply pressure to Ben Harosh to incriminate the Abergil brothers but failed.
Hai Vaknin, called an " Abergil henchman " by the Israeli daily Haaretz, was arrested in the USA in 2006. In January 2011, he signed a plea bargain, confessing to money laundering and receiving a 57-month jail sentence, which he had already served, and was ordered to serve three years under supervised release. His description of loans and extortion practices was expected to help convict Itzhak and Meir Abergil.
In early August 2008, Itzhak and Meir Abergil were arrested on suspicion of involvement in the murder of Margarita Lautin who had died after being mistakenly shot during a failed assassination attempt by members of the Abergil mob.
On August 26, 2008 Itzhak and Meir Abergil along with Moshe Malul and Israel Ozifa were brought before a Jerusalem Magistrate ' s Court judge for their alleged role in the killing of Israeli drug dealer Sami Atias in Encino in August 2003, as a revenge for his allegedly having stolen money from them. They were remanded in custody together with Sason Barashi as a result of a request by law enforcement in the United States for their extradition. The indictment includes four different crimes that are attributed to Yitzhak Abergil: involvement together with Malul in the murder of Atias in California in 2003, trade in Ecstasy, extortion and violence against businessmen, and money laundering and fraud. The inquiry into the case had lasted for six years, involving the FBI, tax authorities and law enforcement officials in more than ten countries in America, Europe, Asia and the Middle East.
In July 2009, a Jerusalem District Court approved the state ' s request to extradite the Abergils, Sasson Barashy, Moshe Malul and Israel Ozifa to the US. In December 2010, the Supreme Court rejected an appeal by the three associates of the Abergil brothers.
On January 12, 2011 Itzhak and Meir Abergil, together with

Figure 9: **Qualitative Analysis for GPT-2-small.** Comparison of the most influential training example retrieved by the baseline EKFAC and our weighted EKFAC. The weighted method retrieves a training example that is semantically more relevant to the query.

**Query**

is often more sophisticated than that used by other eco-extremist groups.

In summary, while the ALF has a history of violent direct action, they have also employed tactics like sabotage, property damage, and animal liberation to achieve their goals. The group's modus operandi includes reconnaissance, surveillance, and careful planning before carrying out an operation. Their actions are typically targeted at specific industries or individuals perceived as responsible for harm to animals or the environment. The ALF's willingness to use violence, combined with their resourcefulness and ability to adapt, makes them a significant threat to law enforcement and private security agencies.

### Earth Liberation Front (ELF)

The ELF is another eco-extremist group known for its radical anti-environmental activities. Founded in the 1990s, the ELF is believed to be a spin-off of the ALF, but some experts argue that they may be separate entities. Like the ALF, the ELF advocates for the protection of the environment and wildlife through direct action, including arson, sabotage, and property destruction.

The ELF has been linked to numerous incidents across North America, Europe, and Australia, including arsons, bombings, and vandalism targeting industries such as logging, mining, and biotechnology. Some notable examples include:

1. August 2001: An ELF cell claimed responsibility for setting fire to a University of Washington research center, which housed a genetic engineering lab.
2. October 2002: A suspected ELF cell set ablaze a timber company's office building in Oregon, causing $10 million in damages.
3. June 2015: An ELF cell was accused of torching a fossil fuel pipeline construction site in Pennsylvania.

Like the ALF, the ELF uses various tactics to evade detection, including using coded language, encrypting messages, and employing secure communication channels. They often claim responsibility for their actions through anonymous communiques sent to media outlets and online forums.

Key characteristics of ELF operations include:

* Targeted industry disruption: The ELF focuses on disrupting specific industries deemed harmful to the environment, such as logging, mining, and fossil fuels.
* Use of violence and intimidation: The ELF employs violent means, such as arson and bombing, to convey their message and

**Top-1 Retrieved by Baseline LoGRA**

a had left her crutches at home that evening, because her friends were able to help her walk. Her legs have been partially paralyzed since Jan. 25, 2014, the third anniversary of the revolution, when security forces shot her in the back during a demonstration.She disappeared on that evening, along with her two friends. When they left the restaurant, several police officers dressed in civilian clothing pulled them into a minivan.The family had no idea what had happened, and there was no news from her for two weeks. Esraa's father went on television to ask the public for help. They contacted an attorney, who filed a complaint with the attorney general. "Tell me where she is and we will investigate the case," the general prosecutor's office replied, suggesting that perhaps she had run off with a lover.It was only by accident that the family learned what had happened to Esraa. A young woman who had seen her in prison contacted the family via Facebook.Perfidious TacticsDuaa El-Taweel says she talks in her sleep at night, constantly repeating the number of her sister's missing person report: 1191. The day before our interview, she saw her sister again for the first time since her disappearance, but only from a distance, beyond a barrier in front of the building that houses the prosecutor general's office. Esraa's detention was extended by 14 days, but the same thing has been happening every two weeks. She is charged with being a member of the Muslim Brotherhood, and of having disseminated false reports and provided information to other countries."Those are the top three charges if you want to arrest someone these days," says her attorney, Halem Henish. El-Taweel denies them all. Henish describes the enforced disappearances as a perfidious tactic, because "it allows the government to hide people from the law." A person who is officially arrested cannot be interrogated without an attorney present. The person's case would have to be presented to the public prosecutor within 24 hours, and the government would have to release him or her if there were no indictment, says Henish.But the "disappeared" are stuck in a legal vacuum of sorts. They are initially taken to a building owned by the state security service. "Confessions are forcibly extracted from them there," sometimes through torture, Henish explains. Then an indictment is prepared. According to her attorney, Esraa El-Taweel was interrogated for 18 hours, and

**Top-1 Retrieved by Weighted LoGRA (Ours)**

in the wake of past attacks and many ALF operatives have diverted their efforts toward the homes of executives and researchers (like the UC-Santa Cruz researchers) and other soft targets. Gravitating toward softer targets makes it less likely operatives will be caught in the act. Additionally, the surveillance tradecraft utilized by the ALF and its operatives and the operational security they practice is usually better than that demonstrated by jihadist lone wolves. Organizations such as the Ruckus Society conduct detailed courses on preoperational surveillance, which is called "scouting" in their parlance. Also, since ELF/ALF activists tend to be young Caucasians, they are generally not viewed as a potential threat, even if they are spotted conducting surveillance. Moreover, since these activists have focused mainly on attacks that cause property damage, law enforcement has understandably not placed the same priority on catching ELF/ALF activists as it has other actors such as jihadists, who intentionally target people. In Bond's case, he might have had some difficulty not drawing attention to himself as he cased leather stores and foie gras restaurants because he had tattoos covering half his face with the word "vegan" tattooed across his throat in large block letters flanked on either side by crossed wrenches. "Monkey wrenching" is a term widely used by activists associated with ELF/ALF and anarchist groups to refer to direct-action attacks that involve property destruction such as arson. Anyone involved in animal research or selling animal products who is observant enough would surely look suspiciously upon a person with such distinctive markings. When all of these factors combine, it is usually very difficult to solve an ELF/ALF arson or bombing case unless a mistake is made, or a confidential informant comes forward. Most successful prosecutions in such cases have come as a result of informants, and because of this we have witnessed a cat-and-mouse game between activists and the government regarding informants, with activist groups pressing informants to commit illegal activities before being accepted and the government giving them permission to do so. Although the CI in the Bond case was just an acquaintance of Bond who was concerned about his arson activities and not a person specifically dispatched to penetrate the movement, without the help of the CI, the government probably had very little chance of identifying Bond. Animal rights blogs and websites have already begun dissecting the Bond case and providing lessons learned to current (and aspiring) animal rights activists. Many of these sites have focused on Bond's mistake in confiding in the CI and have indicated that they believe the informant

Figure 10: **Qualitative Analysis for Llama3-8B-Instruct.** Comparison of the most influential training example retrieved by the baseline LoGRA and our weighted LoGRA. The weighted method retrieves a training example that is semantically more relevant to the query.

**Query**

hoping to improve their chances of winning a World Series title. The Red Sox did just that before the 2013 season, adding Shane Victorino, Mike Napoli, and Jonny Gomes to an already strong core. They won the division and eventually the World Series. But other teams made similar moves and didn't have the same success. The Yankees signed Kevin Youkilis, Travis Hafner, and Vernon Wells, but they missed the playoffs entirely. The Dodgers traded for Hanley Ramirez and Adrian Gonzalez, but they were eliminated in the National League Championship Series. There's no guarantee that making big moves will lead to a successful season.
2. The importance of starting pitching can't be overstated.Starting pitchers are often the most critical component of a team's rotation. The Red Sox had a strong group last year, with Clay Buchholz, Jon Lester, and John Lackey all having solid seasons. The Tigers' Justin Verlander and Max Scherzer formed one of the best 1-2 punches in baseball, while the Cardinals' Adam Wainwright and Lance Lynn were equally impressive. A strong starting rotation can carry a team through tough stretches and give them a chance to win in the postseason.
3. Bullpens are becoming increasingly important.In recent years, bullpens have become more specialized and crucial to a team's success. The Red Sox's bullpen was a strength throughout the season, with closers Koji Uehara and Craig Breslow being key contributors. Other teams like the Pirates and Athletics also relied heavily on their 'pen to get outs and secure victories. As the game becomes more focused on matchups and leverage situations, the role of the relief pitcher is only going to continue to grow.
4. Defense matters, especially in the postseason.The Cardinals' defense was a major factor in their World Series victory, as they turned double plays and made crucial plays in the field to snuff out rallies. The Tigers' defense was similarly stingy, with Miguel Cabrera and Prince Fielder forming a formidable 3-4 combination at the plate. In the postseason, every play counts, and a team's ability to defend its territory can be the difference between advancing or going home

**Top-1 Retrieved by Baseline LoGRA**

of how machine learning algorithms work can help provide understanding as to what sort of questions machine learning can help to answer, and what sort of questions are problematic.Secondly, I hope this series encourages some of you to dig deeper, to learn more about this topic. Machine learning is a rapidly growing field that is expanding to every aspect of life. This includes, recommendation engines on websites, astronomy – where it helps to identify stars and planets, the pharmaceutical industry – where it is being used to predict which molecular structures that are likely to produce useful drugs, and maybe most famously, in training self-driving cars to drive in the real world. Whatever your primary interest, there is likely to be some machine learning applications being developed or being used already.[1] There are a range of metrics that can be used to do this. For available metrics in the Scikit Learn package, see here.Full script:import pandas as pd import numpy as np import xgboost as xgb from sklearn import cross_validation, decomposition, grid_search from sklearn.preprocessing import LabelEncoder
########################################### # Functions #
########################################### #
Remove outliers def remove_outliers(df, column, min_val, max_val): col_values = df[column].values df[column] = np.where(np.logical_or(col_values<=min_val, col_values>=max_val), np.NaN, col_values) return df # Home made One Hot Encoder def convert_to_binary(df, column_to_convert): categories = list(df[column_to_convert].drop_duplicates()) for category in categories: cat_name = str(category).replace(" ", "_").replace("(", "").replace(")", "").replace("/", "_").replace("-", "").lower() col_name = column_to_convert[:5] + '_' + cat_name[:10] df[col_name] = 0 df.loc[(df[column_to_convert] == category), col_name] = 1 return df # Count occurrences of value in a column def convert_to_counts(df, id_col, column_to_convert): id_list = df[id_col].drop_duplicates() df_counts = df.loc[:,[id_col, column_to_convert]] df_counts['count'] = 1 df_counts = df_counts.groupby(by=[id_col, column_to_convert], as_index=False, sort=False).sum() new_df = df_counts.pivot(index=id_col, columns=column_to_convert, values='count') new_df = new_df.fillna(0) # Rename Columns categories = list(column_to_convert].drop_duplicates()) for category in categories: cat_name = str(category).replace(" ",

**Top-1 Retrieved by Weighted LoGRA (Ours)**

When it's all over, we media know-it-alls tally everything up, then declare with breathtaking confidence who won and who lost. The problem is, we're doing it wrong. Signing the highest-priced free agent is great, but it might not get you far if you neglect other roster weaknesses to get there.The 2013 Angels were a perfect example of this phenomenon. When the Halos spent $125 million on Josh Hamilton, it was hard to look at that team and see anything other than Mike Trout, Albert Pujols, and Hamilton laying waste to American League pitching. Problem was, their own pitching stunk. The top of the rotation was functional, if nothing special, with C.J. Wilson, 24 starts from Jered Weaver, and 24 more from Jason Vargas. But the 58 starts made by Jerome Williams, Tommy Hanson, and especially Joe Blanton were a nuclear disaster, and the bullpen was the sixth-worst in baseball by fielding-independent measures. This isn't a one-year trend, either. One year earlier, the Angels (along with the Marlins) landed some of the biggest names on the market (Pujols, Wilson, Jose Reyes, Mark Buehrle) only to disappoint the following season.2. Bargain hunting can and does often pay off.Francisco Liriano. Russell Martin. Bartolo Colon. James Loney. Koji Uehara. Marlon Byrd. The list of low-priced free agents who delivered big returns for their teams in 2013 is a long one. Each of these players was acquired cheaply because he had a perceived defect. The Pirates got Martin cheap because he hit just.211 in 2012 — even though he bopped 21 homers, played his usual strong defense, and was just 29 years old. Loney hit a catastrophic.249/.293/.336 in 2012, but he was just 28 years old, played excellent defense, and had put up some decent offensive seasons, albeit without the power you'd hope for from a slugging-heavy position like first base. Liriano and Uehara had injury histories, though both had shown flashes of

Figure 11: **Qualitative Analysis for Llama3-8B-Instruct.** Comparison of the most influential training example retrieved by the baseline LoGRA and our weighted LoGRA. The weighted method retrieves a training example that is semantically more relevant to the query.

