# OpenReview forum: "Learning to Weight Parameters for Training Data Attribution"
_ICLR.cc/2026/Conference — ICLR 2026 Poster_

### Official Review · Reviewer_4e32 · 2025-10-29

**Soundness:** 2
**Presentation:** 3
**Contribution:** 3
**Rating:** 4
**Confidence:** 3

**Summary:**

Many methods for data attribution consider model parameters uniformly, as if all parameters contribute equally. While influence functions do consider how some directions in parameter-space matter more than others, the true Hessian is intractable in practice and only subsets of parameters are used. The authors show that attribution quality is not uniform across parameters, and propose a method that learns parameter group weights from data.

**Strengths:**

To my knowledge, this work's exploration of how attribution is impacted by the heterogeneous nature of parameters (that some parameters are more important than others for particular tasks) is the first of its kind. It brings up some new findings, which concur with pruning work:
1. up blocks exhibit higher LDS attribution scores than others. In pruning, deeper computation is also more high-level/hierarchical
2. there is significant variation in terms of how much each parameter (and even each parameter group) affect attribution. Without variation in parameter importance in the context of pruning, we wouldn't be able to prune large fractions without affecting accuracy proportionally.

Separately, using weights from section 4.2, their method improves attribution (LDS score) of TracIn and TRAK for CNN and ViT. On text data (WikiText-103 w/ GPT-2-small) LDS score also improves for all methods tested (TracIn, TRAK, LoGRA, EKFAC).

Appreciate the extra detail in appendix motivating Eq. 6, which describes how minimizing their loss equates to maximizing the SNR of the attribution score of the existing base attribution method.

**Weaknesses:**

Main issue: The authors do not specify (or at least do not make clear) what data is used to learn the weights from section 4.2. In section 5.1 they state "weights are learned using samples that are not in the training or test data", but what are that data? In the first imagenet section 5.1.1, there is no specification as to the data that is used, and that continues for the following sections on other data modalities. More specifically, in section 5.1.1, lines 372-376 are on how they train the models and how they evaluate LDS. Lines 388-396 are about evaluating mislabeled data detection. This section misses the most important part: what data is used for $w$ (the column that brings the improvement)? Without information on the above, the improvement could be due to a bit of metric hacking of LDS score.

Second issue: The section on Finegrained data attribution focuses solely on Recall@10. Under this task, I feel simple baselines like using feature embeddings from pretrained models [1] should be tried. Prior work like Datamodels also do this and call it "representation distance" (See Section F.3.1 Estimation using baselines of [2])

[1] Singla et al., 2023. "A Simple and Efficient Baseline for Data Attribution on Images" (https://arxiv.org/abs/2311.03386)

[2] Ilyas et al., 2022. "Datamodels: Predicting Predictions from Training Data" (https://arxiv.org/abs/2202.00622)

Willing to raise score if weaknesses are corrected and questions addressed.

**Questions:**

Is maximizing the SNR of an attribution score a form of "metric hacking" of the LDS score? Why or why not? As a simple example, suppose we were measuring RMSE for some task. If one wanted to get a lower one could optimize to reduce variance and this would produce lower RMSE but not necessarily be better at the task (this is what I mean by metric hacking)

---

> ### Author Response · Authors · 2025-11-21
>
> We thank the reviewer for their careful reading and for highlighting critical points about evaluation and potential metric hacking. We appreciate that they view our exploration of parameter heterogeneity as novel and are willing to raise their score if the concerns are addressed. We respond in detail below.
>
> ### 1. Data used to learn the weights and separation from evaluation data
>
> > Main issue: The authors do not specify (or at least do not make clear) what data is used to learn the weights from section 4.2. [...] This section misses the most important part: what data is used for $w$? Without information on the above, the improvement could be due to a bit of metric hacking of LDS score.
>
> We apologize for the lack of clarity in the original main text. The revised manuscript now explicitly specifies, for each setting, **three disjoint roles** for data:
>
> - **Model training data** (for learning model parameters $\theta$).
> - **Weight-learning data** (for optimizing parameter-group weights $w$).
> - **Evaluation data** (for LDS, tail-patch, and fine-grained Recall@10).
>
> Concretely:
>
> - **Image classification (ImageNet, ResNet‑18 / ViT‑B/16).**
>   - Models are trained on a 50,000-sample subset of the training split of ImageNet. For the mislabeled data detection experiment, we corrupt 10% of the labels on this subset.
>   - Weights $w$ are learned using 1,000 additional images sampled from the training split of ImageNet, disjoint from the 50,000-sample subset.
>   - LDS is evaluated on another 1,000 images sampled from the validation split of ImageNet, disjoint from both the model-training set and the weight-learning set.
> - **Language modeling (WikiText‑103, GPT‑2-small).**
>   - The model is trained on the WikiText‑103 training split, containing 232,585 sequences of length 512.
>   - Weights $w$ are learned using 523 sequences from the test split.
>   - LDS is evaluated on the 487 sequences from the validation split, and tail-patch is evaluated on the same held-out queries.
> - **Diffusion models (CIFAR‑2, ArtBench‑2, Naruto, SB‑Pokemon).**
>   - Models are fine-tuned on the respective training sets, detailed in Appendix B.1.
>   - Weights $w$ are learned using 1,000 query images generated by the trained model using the prompts described in Appendix B.1. For fine-grained attribution, instead, we only use 200 query images generated with prompts containing only concepts from the train split (detailed in Section 5.2 and Appendix B.1).
>   - LDS is evaluated using 1,000 query images generated by the trained model using the same prompting schemes, but with disjoint sets of initial random seeds from the weight-learning set. For fine-grained attribution, for both the train and validation splits, there are 500 query images containing only concepts from that split, generated with different initial random seeds.
>
> In summary, **the data used for model training, weight learning, and evaluation are all disjoint**, so improvements in LDS cannot be attributed to reusing evaluation data for weight learning. We have made this clear in the revised Section 5, and added a table in the appendix summarizing the data used for each experiment.

---

> ### Author Response · Authors · 2025-11-21
>
> ### 2. Is maximizing SNR a form of "metric hacking" LDS?
>
> > Is maximizing the SNR of an attribution score a form of "metric hacking" of the LDS score? [...] If one wanted to get a lower RMSE one could optimize to reduce variance and this would produce lower RMSE but not necessarily be better at the task.
>
> We appreciate this important question. Our view is that our objective does **not** constitute metric hacking of LDS, for several reasons:
>
> 1. **Different objectives.** Our loss operates only on the raw attribution scores $\widetilde{\tau}(x_q, x_n; w)$: it selects the top-$k$ training examples by attribution score (not by LDS) and maximizes their average while controlling the overall $\ell_2$ scale. LDS, by contrast, is the Spearman correlation between true subset effects and attribution-based predictions over many retrained models. LDS values are never used, directly or indirectly, in the optimization of $w$.
> 2. **Evidence beyond LDS.** The same learned weights improve several other evaluations that are not defined in terms of LDS, including mislabeled-data AUC and tail-patch score. They also yield more meaningful qualitative behaviour: in diffusion models, Figure 4 shows that weighted attribution surfaces training images that better match the target concept, and in the revised Appendix C.7 we include analogous qualitative examples for GPT‑2-small and Llama3‑8B-Instruct, where weighted attribution retrieves training sequences that more clearly explain the model’s predictions.
> 3. **Ablations corroborate the SNR view.** If our loss were merely "hacking" LDS through simple score transformations (for example, maximizing the gap between top and bottom scores), then other straightforward variants should perform similarly or better. Instead, we observe the opposite: removing the normalization term ("No-Norm" in Table 9) causes performance to collapse, and a "Top‑$k$ minus Bottom‑$k$" objective does not outperform our main loss. These behaviours are exactly what our SNR derivation predicts (the denominator must track noise, and the bottom tail is noisy because highly harmful influencers are rare), and are hard to explain as superficial metric hacking.
>
> Intuitively, the loss encourages the attribution mechanism to **rely more on parameter groups whose scores carry a stable influence signal and to down-weight groups whose scores behave like noise**, rather than manipulating the metric directly. This leads to more accurate influence rankings, as reflected by the consistent improvements across LDS, mislabeled‑data detection, tail‑patch scores, and qualitative analyses. We have added a short paragraph to Appendix A explicitly addressing this concern.
>
> ### 3. Fine-grained baselines: representation distance
>
> > The section on Finegrained data attribution focuses solely on Recall@10. Under this task, I feel simple baselines like using feature embeddings from pretrained models [Singla et al., Datamodels] should be tried.
>
> We thank the reviewer for this suggestion and for pointing us to the relevant work.
>
> We agree that evaluating fine-grained attribution with a single metric can be weak, but since our work is, to our knowledge, the first to propose fine-grained data attribution, we view this as a reasonable starting point and expect that more refined metrics and benchmarks will be developed in future work.
>
> Following the reviewer’s advice, we have added a **representation-distance baseline** in the revised manuscript: using the same MoCo-pretrained ResNet-50 as in [1], we rank training images by feature distance to the query and report Recall@10 for subject, style, and background on SB‑Pokemon. The results, now included in Table 6, show that the D-TRAK baseline and D-TRAK with learned weights outperform the representation baseline on each target concept.
>
> [1] Singla et al., 2023. "A Simple and Efficient Baseline for Data Attribution on Images" (https://arxiv.org/abs/2311.03386)
>
> ---
> We thank the reviewer again for their thorough review that helps us to further clarify our experimental setup and baselines. We believe the manuscript is now substantially clearer and more informative thanks to your suggestions, and we would be very happy to further clarify any remaining concerns in future iterations.

---

### Official Review · Reviewer_tnbV · 2025-10-31

**Soundness:** 2
**Presentation:** 3
**Contribution:** 3
**Rating:** 6
**Confidence:** 3

**Summary:**

The paper is motivated by the observation that network parameters are not uniformly contribute to the model behavior, and thus the data attribution process. The paper propose a novel way to choose the weight of each parameter. The new method shows good effectiveness on traditional evaluation methods as well as provide more fine-grained attribution of concepts.

**Strengths:**

- The paper propose an interesting question (and opportunity) to improve the effectiveness of data attribution. This direction is discussed but not checked in depth in previous literature.
- The self-supervised mechanism to learn parameter weight is practical and easy to use.
- The improvement over standard data attribution methods are obvious.

**Weaknesses:**

- The main weakness lies in the self-supervised weight learning loss design. The analysis use a signal-to-noise ratio model to see the attribution score and try to optimize the parameter weight to get highest signal-to-noise.
  - Problem is that the definition of signal is related to the top-k attribution score. The decision is not justified well in Section 4.2 as well as in Appendix A.
  - Intuitively, top-k attribution score is very important (to the counterfactual prediction), while the least-k attribution score is also important and affect the counterfactual prediction. Why only choose top-k could be stated more carefully in the paper.

- The fine-grained attribution of concepts is somehow weak in both motivation, method design and experiment results
  - In Section 3.3, the motivation of revising parameter weight for concept attribution use similarity as the groundtruth, which is not a good practice for data attribution. Since how does the model learn a data may be different from the intuition that "similar data helps most".
  - The method design relies on the query data's distribution, could this also work for uniform/hessian based parameter reweighing?
  - Figure 4 shows some "imperfect" attribution results: e.g., style No.5 (not in the watercolor style), background No.2&5 (not with the same background)...

**Questions:**

- A core question could be: what is the essential reason that makes the hessian based parameter reweighing does not work well?

---

> ### Author Response · Authors · 2025-11-21
>
> We thank the reviewer for their detailed and thoughtful comments. We appreciate that they found the question of parameter heterogeneity compelling, and that they view our self-supervised mechanism as practical and effective. Below we address the concerns around the loss design and fine-grained attribution.
>
> ### 1. Top‑k/SNR self-supervised loss
>
> > The main weakness lies in the self-supervised weight learning loss design. The analysis use a signal-to-noise ratio model to see the attribution score and try to optimize the parameter weight to get highest signal-to-noise. Problem is that the definition of signal is related to the top-k attribution score. The decision is not justified well in Section 4.2 as well as in Appendix A. Why only choose top-k could be stated more carefully in the paper.
>
> We appreciate this concern and agree that the role of top‑$k$ (and the omission of bottom‑$k$) should be stated more carefully. At a high level, our SNR model assumes that the **largest-magnitude scores contain most of the usable signal**, whereas the bulk of scores are near zero and noise-dominated. The top‑$k$ part of our loss is therefore intended to focus on where the base method has the highest chance of surfacing truly influential examples, without claiming that harmful (negative) influences are unimportant in general.
>
> To test the reviewer’s intuition about the bottom‑$k$, we added ablations that directly use least‑$k$ scores (Table 9 in the appendix). We considered both a ***Top‑$k$ minus Bottom‑$k$*** objective and a ***Bottom‑$k$-only*** objective (newly added in the revised manuscript). In practice, both variants hurt performance: *Top‑$k$ minus Bottom‑$k$* is consistently slightly worse than our *Top‑$k$* loss, and the *Bottom‑$k$-only* loss provides less improvement and in one case even worse than the unweighted baseline. A natural explanation is that truly harmful training examples with large negative influence are rare, while the vast majority of points have near-zero true influence; neural networks tend to "memorize" specific examples but does not actively "forget" the exact opposite of a point.
> Therefore, we choose the simpler *Top‑$k$* formulation in the main paper: it is more stable, avoids introducing an extra hyperparameter for bottom‑$k$, and empirically yields the best results.
>
> ### 2. Fine-grained attribution: motivation, evaluation, and results
>
> > The fine-grained attribution of concepts is somehow weak in both motivation, method design and experiment results. In Section 3.3, the motivation of revising parameter weight for concept attribution use similarity as the groundtruth, which is not a good practice for data attribution. Since how does the model learn a data may be different from the intuition that "similar data helps most". Figure 4 shows some "imperfect" attribution results.
>
> Our motivation for fine-grained attribution is to go beyond a single "overall" influence score and provide **concept-specific, interpretable attribution**. In many applications a user cares about only part of the model’s behaviour for a given query, for example the artistic style of a generated image. Learning separate weights for subject, style, and background allows us to highlight which training examples and which parameter groups are most important for each semantic aspect, and gives a new way to understand **which parts of the network implement which functionality**.
>
> Regarding the evaluation protocol, we are not using feature "similarity" as ground truth. In the SB‑Pokemon setting we construct a controlled fine-tuning dataset where, for each concept (for example, a specific style), the only supervision the model ever receives for that concept comes from the images in its corresponding training category. Under this construction, the examples in that category are precisely the training points that supply the signal for that concept, so we treat them as the ground-truth contributor set and measure the fraction of the top-attributed examples that are in the set. We agree that this recall metric is somewhat coarse and does not capture all nuances of influence. Since our work is, to our knowledge, the first to propose fine-grained data attribution, we view this as a reasonable first evaluation and expect that more refined metrics and benchmarks will be developed in future work.
>
> As for the "imperfect" qualitative results in Figure 4, this behaviour is in fact expected given our design. Our method up-weights layers that are helpful for a target concept and down-weights others, but it cannot completely remove all signal from non-target concepts. As a result, some proponents that are relevant to other concepts can still appear among the top-ranked examples, even though target-concept contributors are much more prominent than in the unweighted baseline. We have added a remark to clarify this point in the discussion of Figure 4.

---

> > ### Author Response · Authors · 2025-11-21
> >
> > ### 3. Dependence on the query distribution and applicability to uniform/Hessian-based reweighting
> >
> > > The method design relies on the query data's distribution, could this also work for uniform/hessian based parameter reweighing?
> >
> > Our method does use a **query distribution $\mathcal{Q}$** during weight learning, but the learned weights $w$ are global rather than query-specific. In practice we sample $\mathcal{Q}$ to reflect the behaviours we care about (for example, general outputs or style-focused outputs), then learn a single $w$ that works across many queries. Our generalization experiments (Appendix C.4) show that weights learned under one dataset often transfer well to others, which suggests that they capture stable architecture-level patterns rather than overfitting to a particular choice of $\mathcal{Q}$.
> >
> > Regarding the connection to uniform or Hessian-based parameter reweighting, our formulation is **agnostic to how the base score is computed** and can be applied on top of both uniform-gradient methods (TracIn) and Hessian-based methods (TRAK, EKFAC, D-TRAK, etc.) One can view classical Hessian-based reweighting as providing an implicit, curvature-driven set of weights, and our approach as learning an additional, explicit set of group-wise weights from attribution quality. In this sense we already extend uniform and Hessian-based schemes rather than being incompatible with them.
> >
> >
> > ### 4. Why implicit Hessian-based reweighting does not work well
> >
> > > A core question could be: what is the essential reason that makes the hessian based parameter reweighing does not work well?
> >
> > As discussed in Section 3.1, practical influence-function methods must approximate the Hessian using random projections and Gauss-Newton/K-FAC. These approximations are inherently lossy and yield only imperfect scaling by local curvature. Recent work has also shown that even in small MLPs, these Hessian approximations can be substantially inaccurate and unfaithful to the true Hessian[1]. This leaves room for a complementary mechanism that does not rely on the same approximations but directly learns the weights from attribution results.
> >
> > [1] Hong, Dat Minh, et al. "Better Hessians Matter: Studying the Impact of Curvature Approximations in Influence Functions." Mechanistic Interpretability Workshop at NeurIPS 2025.
> >
> > ---
> > We thank the reviewer for raising these important points. We hope that our detailed ablations on the loss function and the new fine-grained baselines help clarify the motivation and effectiveness of our approach. If any aspects of your concerns remain unaddressed or we have not fully captured their intent, we would very much welcome further clarification in order to improve the work.

---

### Official Review · Reviewer_uSNX · 2025-10-31

**Soundness:** 3
**Presentation:** 3
**Contribution:** 3
**Rating:** 6
**Confidence:** 3

**Summary:**

This paper addresses the problem of data attribution, which aims to identify the training samples most responsible for a model’s prediction. While existing gradient-based methods (e.g., TracIn, TRAK, Influence Functions) treat all parameters equally, this paper observes that attribution quality varies significantly across different parameter groups.   To tackle this, the authors propose a learnable parameter-weighted attribution framework, introducing explicit, non-negative group weights to scale gradient contributions. Extensive experiments on image classification (ResNet-18, ViT-B/16), language modeling (GPT-2), and diffusion models (Stable Diffusion, D-TRAK, DAS) show consistent improvements across metrics like LDS, mislabeled-data AUC, and tail-patch score.

**Strengths:**

1. Identifying parameter heterogeneity as a fundamental but overlooked factor in data attribution.
2. Generalizes across TracIn, TRAK, DAS, and others with a single weighting formulation.
3. Strong, consistent results across image, text.
4. Learned weights provide semantic insights into layer-level specialization

**Weaknesses:**

1. Technically, TRAK and similar Hessian-based methods already introduce an implicit parameter weighting through the approximation of $H^{-1}$, which scales gradients by local curvature. The proposed explicit weighting can thus be viewed as learning functional heterogeneity on top of curvature-based scaling.

2. The self-supervised loss bootstraps from existing methods (e.g., TRAK), so its ultimate accuracy may inherit their biases.

3.  Equation (6) defines the weighted attribution as
  $\tilde{\tau}(x_q, x_i; w) = g(x_q)^\top \mathrm{Diag}(w)K g(x_i),$ but shouldn't it be $\tilde{\tau}(x_q, x_i; w) = g(x_q)^\top \mathrm{Diag}(w^2) Kg(x_i),$? which corresponds to reweighting both query and training gradients equally.

**Questions:**

1. How sensitive is the learned weighting to the quality of the initial pseudo-ranking? Would a noisy baseline still lead to meaningful weights?

---

> ### Author Response · Authors · 2025-11-21
>
> We thank the reviewer for their constructive and encouraging review, and for highlighting the importance of parameter heterogeneity and the semantic interpretability of our learned weights. Below, we address your concerns in turn.
>
> ### 1. Relation to Hessian-based / curvature-induced weighting
>
> > Technically, TRAK and similar Hessian-based methods already introduce an implicit parameter weighting through the approximation of $H^{-1}$, which scales gradients by local curvature. The proposed explicit weighting can thus be viewed as learning functional heterogeneity on top of curvature-based scaling.
>
> We fully agree that TRAK and similar methods implicitly weight directions in parameter space via approximate curvature. Our goal is not to replace this mechanism, but to **complement** it in two key ways:
>
> 1. **Quality of the curvature signal.** As discussed in Section 3.1, practical influence-function methods approximate the Hessian using damped Gauss–Newton or K‑FAC, under simplifying assumptions about the model at a local optimum. These approximations are inherently lossy and yield only imperfect scaling by local curvature. Recent work has also shown that even in small MLPs, these Hessian approximations can be substantially inaccurate and unfaithful to the true Hessian[1]. This leaves room for a complementary mechanism that does not rely on the same approximations but instead learns the weights directly from attribution results.
> 2. **Level of granularity and interpretability.** Curvature-based methods do not provide explicit, interpretable weights per architectural component. Our approach learns **explicit group-wise weights (for example, per layer or block)**, making the heterogeneity directly inspectable and enabling semantic specializations (such as subject versus style).
>
> Empirically, this complementarity is evident: our explicit weights improve attribution quality even for Hessian-based methods, while providing more significant increase for the non-Hessian-based method, TracIn. This suggests that curvature-based weighting alone does not fully capture the functional and semantic heterogeneity that our method targets, and that learning explicit group-wise weights from attribution quality provides additional value.
>
> [1] Hong, Dat Minh, et al. "Better Hessians Matter: Studying the Impact of Curvature Approximations in Influence Functions." Mechanistic Interpretability Workshop at NeurIPS 2025.
>
> ### 2. Bootstrapping from existing methods and potential bias
>
> > The self-supervised loss bootstraps from existing methods (e.g., TRAK), so its ultimate accuracy may inherit their biases.
>
> We agree that any self-supervised approach that bootstraps from existing attribution methods will inevitably reflect some of their biases, and we have made this limitation explicit in Conclusion (Section 6) in the revised manuscript.
>
> To explore alternatives, we also implemented a **supervised variant** that does not rely on the base ranking: it uses augmented views of a training example as pseudo-positives, as described in Appendix C.3 (Supervised Loss with Data Augmentation). This formulation can in principle reduce dependence on the base method, although it introduces its own dependence on manually designed augmentations. Empirically, it underperforms our self-supervised top‑$k$ loss across methods (see Table 9), which is why we retain the latter as our main objective.
>
> Overall, we view inherited bias from existing attribution methods as an acceptable trade-off for achieving robust empirical gains without requiring ground-truth attribution labels.

---

> > ### Author Response · Authors · 2025-11-21
> >
> > ### 3. Equation (6): weighting the query vs both query and training gradients
> >
> > > Equation (6) defines the weighted attribution as $g(x_q)^\top \mathrm{Diag}(w) K g(x_i)$, but shouldn't it be $g(x_q)^\top \mathrm{Diag}(w^2) K g(x_i)$, which corresponds to reweighting both query and training gradients equally?
> >
> > We apologize for the confusion and appreciate the opportunity to clarify this design choice.
> >
> > - For the **special case $K = I$** (e.g., TracIn-style dot products), weighting either the query or the training gradient—or both—can be reparameterized, since a global scale on one side can be absorbed into $w$. In this setting, our formulation is equivalent to symmetric weighting up to a change of variables.
> > - For **general kernels**, the training-side term $K g(x_i)$ is precomputed for all training examples to make attribution scalable. Applying weights symmetrically would require recomputing this term every time $w$ is updated, which would make the weight-learning process significantly more expensive.
> >
> > Conceptually, we interpret $w$ as specifying **how we trust different parameter groups when reading the query’s gradient signal**, with the training-side features treated as fixed. We have added a remark to relieve the confusion after Eq. (6). Thank you for the careful reading.
> >
> > ### 4. Sensitivity to noisy attribution scores
> >
> > > How sensitive is the learned weighting to the quality of the initial pseudo-ranking? Would a noisy baseline still lead to meaningful weights?
> >
> > Empirically, our experiments demonstrate that **weights learned from a relatively noisy method (TracIn) still significantly improve LDS and other metrics**. Particularly, for Stable Diffusion on CIFAR-2, the LDS before weighting is only $1.39$ while after weighting it is $8.48$. This suggests that our method is capable of extracting meaningful signal even from poor-quality base rankings.
> >
> > To probe this robustness more directly, we conducted a **noise sensitivity analysis** where we injected Gaussian noise scaled by multiples of the layer-wise standard deviation (scale $s$ ranging from 0.1 to 100) into the attribution scores during weight learning. We observed remarkable stability: for ViT on ImageNet, injecting noise with $s=10.0$ only marginally reduced the performance gain; similarly, for Stable Diffusion on ArtBench-2, performance remained consistent even with added noise ($s=0.3$). These results, which we have plotted in Figure 7 in the revised appendix C.3, confirm that our self-supervised objective is highly resilient to noise.
> >
> > Theoretically, this aligns with our design: the objective only requires the top‑$k$ set to be enriched with true influences on average. We thank the reviewer for this insightful question, which prompted us to rigorously verify the robustness of our approach.
> >
> > ---
> > We thank the reviewer again for their constructive feedback. We hope that our clarifications and the new experiments adequately address your concerns.

---

### Official Review · Reviewer_6Ser · 2025-11-01

**Soundness:** 4
**Presentation:** 4
**Contribution:** 3
**Rating:** 6
**Confidence:** 3

**Summary:**

This paper studies the task of training data attribution using influence functions. Specifically, they analyze the varying effects of attribution across the parameter groups in a neural network. The first part of the paper shows this variance empirically, and the second part focuses on learning weights for individual parameter groups. Authors propose a self-supervised methodology to learn these weights. They use a baseline attribution method to provide pseudo ground-truth training examples.

Experiments span ImageNet classification, image generation and language modeling. Results show that the proposed weighting scheme helps improve attribution across a range of attribution methods (TracIn, TRAK, LoGRA, EKFAC). Additionally, the authors also curate a synthetic Pokemon dataset with images by varying subject, style and background. Results show strong results with attributing style, with moderate (/poor) results with subject and background.

Overall, the idea that different parameter groups have different effects on attribution is intuitive. Prior work makes ad-hoc assumptions, and this paper studies this behavior in a more principled fashion. The main concerns are that the paper mostly explored small-sized models and doesn't include any qualitative analysis on where they find gains over the baseline. For instance, does weighting parameter groups instead of pre-selection help improve attribution in LM generated text?

**Strengths:**

- This work presents a methodology to automatically learn weight model parameter groups on their attribution abilities. Prior work in attribution using influence functions focuses on a predefined set of parameter groups. This work presents a more principled solution for this.
- Experiments include comparison against multiple strong attribution baselines.
- The analysis of involving subject, style and background variations in synthetic dataset shows strong results.

**Weaknesses:**

- The language modeling experiments use GPT-2 small and its unclear to me if the gains will necessarily transfer to larger LMs. As the authors highlight in section 2, recent methods (LoGRA, TrackStar) have improved efficiency. Any reason for not experimenting with larger models in this work?
- The paper could benefit from analyzing its relative performance gains across different baseline attribution methods. For instance, they show strong results with TracIn but moderate results with recent methods.
- The paper could also benefit from qualitative analysis of their method for language modeling experiments; this would be an extension to their analysis on the synthetic Pokemon dataset.

**Questions:**

A few additional questions and comments,

- I would update the lines 218-219 to briefly mention the analysis from C.1. This could strengthen your argument in the main text.
- Why does the proposed method show better gains with some attribution methods? For instance, the gains are higher with TRAK and EKFAC and smaller with LoGRA in Table 2. Do you have a hypothesis for these differences?
- Any reason for not including TrackStar (line 127) in your experiments?

---

> ### Author Response · Authors · 2025-11-21
>
> We thank the reviewer for the thoughtful and positive assessment of our work. We are encouraged that you find our treatment of parameter-group heterogeneity "principled" and "intuitive," and that you value the breadth of our experiments across modalities.
>
> Below, we address your questions and describe the additional experiments conducted to resolve your concerns.
>
> ### 1. Scalability to Larger Models
>
> > The language modeling experiments use GPT-2 small and it's unclear if the gains transfer to larger LMs. As the authors highlight in Section 2, recent methods (LoGRA, TrackStar) have improved efficiency. Any reason for not experimenting with larger models in this work?
>
> In our original submission, we used GPT‑2-small because our primary evaluation metric, the Linear Datamodeling Score (LDS), is computationally intensive and requires retraining hundreds of models on data subsets. Scaling LDS evaluation to models the size of Llama-3 or larger is prohibitively expensive for academic resources.
>
> However, we agree that verifying scalability is meaningful. To address this, we have conducted a new experiment using Llama3‑8B‑Instruct and we employed the tail-patch evaluation. The results of our method, reported below and in the revised Appendix C.6 as well, show consistent improvements over the unweighted baselines, mirroring the patterns we have seen for smaller models.
>
> | Method | Tail‑patch score w/o $w$ | Tail‑patch score with $w$ | Improvement $\Delta$ |
> |--------|---------------------------|---------------------------|--------------------|
> | Random | -0.07                     | –                         | –                  |
> | TracIn | 0.04                     | 0.65                      | 0.61 $\pm$ 0.11    |
> | LoGRA  | 1.78                      | 1.92                      | 0.15 $\pm$ 0.05    |
>
>
> ### 2. Variation in gains across attribution methods
>
> > The paper could benefit from analyzing its relative performance gains across different baseline attribution methods. For instance, they show strong results with TracIn but moderate results with recent methods.
>
> We appreciate this observation and agree that the pattern is worth explaining.
>
> - **TracIn** uses raw gradient dot products and treats all parameters uniformly, so our learned weights act as a strong denoising mechanism: they can sharply down‑weight low-SNR groups and up‑weight a few key blocks, leading to large relative gains.
> - **TRAK, EKFAC, JourneyTRAK, D‑TRAK, DAS, and LoGRA** already incorporate non-uniform scaling through curvature (kernel). In these cases our explicit weights still improve LDS and downstream metrics, but the relative gains are more modest because part of the parameter heterogeneity has already been captured implicitly, although in an imperfect way (see Section 3.1).
>
> In short, the more parameter heterogeneity is already implicitly captured, the less our explicit weighting can improve the attribution signal.
>
>
> ### 3. Qualitative analyses for language modeling
>
> > The paper could also benefit from qualitative analysis of their method for language modeling experiments; this would be an extension to their analysis on the synthetic Pokemon dataset.
>
> We agree and have added a new section C.7 in the revised appendix for qualitative analyses of our method on both **GPT‑2-small and Llama3‑8B‑Instruct**. For each validation query, we compare the top‑1 training example retrieved by the baseline attribution method and by its weighted counterpart when given the same query. Our method consistently surfaces training sequences that are more semantically aligned with the query content than those retrieved by the unweighted baseline.

---

> > ### Author Response · Authors · 2025-11-21
> >
> > ### 4. TrackStar and choice of baselines
> >
> > > Any reason for not including TrackStar (line 127) in your experiments?
> >
> > We did not include TrackStar because their implementation is not publicly available. Re‑implementing it from scratch would have required substantial engineering effort, as TrackStar combines several new design choices and many more nuanced differences than our existing attribution pipeline. We therefore prioritized a broad set of baselines for which mature implementations were available (TracIn, TRAK, LoGRA, EKFAC, JourneyTRAK, D‑TRAK, DAS).
> > Nevertheless, TrackStar mainly refines the gradient computation, projection, and Hessian approximation within each layer, and is conceptually compatible with our parameter-weighting framework; we therefore expect that our method could also improve TrackStar when applied on top of it.
> >
> > ### 5. Referencing the similarity analysis (Appendix C.1)
> >
> > > I would update lines 218–219 to briefly mention the analysis from C.1. This could strengthen your argument in the main text.
> >
> > Thank you for this helpful suggestion. In the revised manuscript, we have briefly expanded on the analysis from Appendix C.1. We changed the relevant sentence from
> >
> > `Furthermore, as we show in Appendix C.1, this pattern of heterogeneity is not an artifact of a single setting but an intrinsic characteristic consistently across different datasets and attribution methods.`
> >
> > to
> >
> > `Furthermore, as we show via a cosine-similarity analysis in Appendix C.1, the pattern of per-group LDS scores and the learned weights is highly consistent across datasets and attribution methods, indicating that this heterogeneity reflects a stable, intrinsic property of the model rather than an artifact of any specific setting.`
> >
> > ---
> > We thank the reviewer again for their support and valuable suggestions, which have significantly strengthened our paper. We hope these clarifications and additional results fully address your concerns.

---

### Author Response · Authors · 2025-11-26

We sincerely thank all reviewers for their constructive feedback. Below, we summarize the key concerns raised and main changes we have made in the revised manuscript.

### Summary of Key Concerns Addressed

**1. Clarity on data splits (Reviewer 4e32)**:
The revised manuscript now explicitly specifies, for each experimental setting, **three disjoint data roles**: model training, weight learning, and evaluation. We have added a summary table (Table 7 in Appendix B.1) to make this separation clear.

**2. Potential metric hacking of LDS (Reviewer 4e32)**:
We clarify that our loss selects the top-$k$ training examples by **attribution score** (the influence of each training example on the query), not by LDS (a Spearman correlation computed over hundreds of retrained models). LDS values are never used in the optimization of $w$. We have provided additional evidence against metric hacking, detailed in our full response and in the revised Appendix A.

**3. Scalability to larger models (Reviewer 6Ser)**:
We have conducted new experiments using **Llama3-8B-Instruct** on OpenWebText with tail-patch evaluation. The results, now reported in Appendix C.6, demonstrate that our method consistently improves attribution quality at scale, mirroring the patterns observed for smaller models.

**4. Justification of the top-$k$ loss design (Reviewer tnbV)**:
We have added new ablations in Table 9, including a **Bottom-$k$-only** objective and a **Top-$k$ minus Bottom-$k$** objective. These ablations empirically validate our design choice: the simpler Top-$k$ formulation is more stable and yields the best results, consistent with our SNR derivation.

**5. Robustness to noisy attribution scores (Reviewer uSNX)**:
We have conducted a **noise sensitivity analysis** (Figure 7 in Appendix C.3) showing that our method remains robust even when attribution scores are perturbed with Gaussian noise up to 10x the layer-wise standard deviation.

**6. Qualitative analysis for language modeling (Reviewer 6Ser)**:
We have added a new section (Appendix C.7) with qualitative examples for both **GPT-2-small and Llama3-8B-Instruct**, showing that weighted attribution retrieves training sequences that are more semantically relevant with the query.

**7. Fine-grained attribution baselines (Reviewer 4e32)**
Following the reviewer's suggestion, we have added a **representation-distance baseline** (using MoCo-pretrained ResNet-50) to Table 6. Our method outperforms this baseline on all fine-grained attribution targets.


### Summary of Main Revisions to the Manuscript

- **Section 5:** Clarified data splits for all experiments.
- **Section 6 (Conclusion):** Added discussion of inherited bias as a limitation.
- **Appendix A:** Added a remark addressing the "metric hacking" concern.
- **Appendix B.1:** Added Table 7 summarizing data usage across experiments.
- **Appendix C.3:** Added noise sensitivity analysis and Figure 7.
- **Appendix C.4 and Table 10:** Added Bottom-$k$ ablation.
- **Appendix C.6:** Added Llama3-8B-Instruct experiment.
- **Appendix C.7:** Added qualitative analysis for language modeling.
- **Eq. (5):** Added clarifying remark on asymmetric weighting.
- **Table 6:** Added representation-distance baseline for fine-grained attribution.

---

We have worked to address each concern thoroughly and hope the revised manuscript resolves all outstanding issues. We would be grateful if reviewers could confirm whether their concerns have been adequately addressed, and we remain happy to provide further clarification on any remaining points.

---

### Author Response · Authors · 2025-12-03
**Note to Area Chair**

We thank the Area Chair for their careful handling of the review process and would like to offer a brief note for consideration when making the final decision.

The sole negative reviewer (Reviewer 4e32, score 4) explicitly stated a willingness to "**raise score if weaknesses are corrected and questions addressed.**" In our revision, we have directly tackled all three of their concerns by clarifying data splits, discussing potential LDS "metric hacking", and adding the requested representation-distance baseline.

While the shortened discussion period prevented a follow-up response from the reviewer, we believe the revision fully resolves their concerns and aligns the manuscript with the positive consensus of the other reviewers.

Thank you for your time and consideration.

Best regards, The Authors

---

### Meta-Review · Area_Chair_GCBX · 2026-01-06

**Summary:**

The paper proposes a method to explicitly learn parameter importance weights for gradient-based data attribution, motivated by the observation that attribution quality varies across network parameters. The reviewers generally appreciated the intuition that addressing parameter heterogeneity can improve attribution performance and the self-supervised nature of the proposed solution.

While Review 4e32 initially raised concerns regarding unclear data splits and the lack of baselines for fine-grained attribution, the authors provided a comprehensive rebuttal that addressed these issues with concrete clarifications and added experiments.

**Reviewer Concerns:**

Review 4e32 initially raised concerns regarding unclear data splits and the lack of baselines for fine-grained attribution, the authors provided a comprehensive rebuttal that addressed these issues with concrete clarifications and added experiments.

**Reviewer Scores:**

Review 4e32 would raise the score if the discussion continues.

---

### Decision · Program_Chairs · 2026-01-26

Accept (Poster)